communications
engineering

# Personalized magnetic tentacles for targeted phototothermal cancer therapy in peripheral lungs

Giovanni Pittiglio [1,2,7✉], James H. Chandler [2,7✉], Tomas da Veiga [2], Zaneta Koszowska [2], Michael Brockdorff [2], Peter Lloyd [2], Katie L. Barry[3], Russell A. Harris[4], James McLaughlan [5,6], Cecilia Pompili[6] & Pietro Valdastri [2]

Lung cancer remains one of the most life-threatening diseases and is currently managed through invasive approaches such as surgery, chemo- or radiotherapy. In this work, we introduce a novel method for the targeted delivery of a therapeutic laser for the treatment of tumors in peripheral areas of the lungs. The approach uses a 2.4 mm diameter, ultra-soft, patient-specific magnetic catheter delivered from the end of a standard bronchoscope to reach the periphery of the lungs. Integrated shape sensing facilitates supervised autonomous full-shape control for precise navigation into the sub-segmental bronchi, and an embedded laser fiber allows for treatment via localized energy delivery. We report the complete navigation of eight primary lumina in the bronchi of an anatomically accurate phantom (developed from computed tomography (CT) data) and successful laser delivery for photothermal ablation. We further evaluate the approach in three diverse branches of excised cadaveric lungs, showing a mean improvement in navigation depth of 37% with less tissue displacement when compared to a standard semi-rigid catheter and navigation depth repeatability across all tests of <1 mm.

---

[1] Department of Cardiovascular Surgery, Boston Children's Hospital, Harvard Medical School, Boston, MA, USA. [2] STORM Lab, Institute of Autonomous Systems and Sensing (IRASS), School of Electronic and Electrical Engineering, University of Leeds, Leeds, UK. [3] Leeds General Infirmary–Leeds Teaching Hospital, University of Leeds, Leeds, UK. [4] Future Manufacturing Processes Research Group, School of Mechanical Engineering, University of Leeds, Leeds, UK. [5] Ultrasound Group, Institute of Autonomous Systems and Sensing (IRASS), School of Electronic and Electrical Engineering, University of Leeds, Leeds, UK. [6] Leeds Institute of Medical Research (LIMR), University of Leeds, Leeds, UK. [7] These authors contributed equally: Giovanni Pittiglio, James H. Chandler. ✉email: giovanni.pittiglio@childrens.harvard.edu; j.h.chandler@leeds.ac.uk

Lung cancer has the highest worldwide cancer mortality rate, with 130,000 deaths projected for 2022 in the US alone[1] (https://seer.cancer.gov/statfacts/html/lungb.html). In early-stage non-small cell lung cancer (NSCLC), which accounts for around 84% of cases, curative surgical intervention is the standard of care. However, surgery is typically highly invasive, necessitating the removal of a large portion of lung tissue which is not suitable for all patients and may impact resulting lung function. Furthermore, although diagnosis as part of lung cancer screening programs has demonstrated survival benefit, it has also highlighted the urgent need to find non-invasive and scalable methods to achieve early diagnosis and therapy[2].

Initial screening via medical imaging techniques such as X-ray and computed tomography (CT) will typically require tissue biopsy to confirm the diagnosis. Biopsy sampling may be performed percutaneously using rigid needles or from inside the airways using a semi-rigid endoscope (bronchoscope) and specialized sampling tools[3]. Sampling via a bronchoscope is beneficial as it removes the need to puncture the pleural membranes; however, it is limited in reach due to the size of the bronchoscope (typically 5–6 mm)[4,5]; allowing access down to the second generation of the bronchi via camera-based surgeon-driven navigation (bronchoscopy). To explore the deeper anatomy, electromagnetic navigation (EMN) can be used, where a pre-bent passive catheter is inserted down the bronchoscope's tool channel and is steered by rotating around its axis from the proximal end. EMN is performed based on a pre-operative CT scan of the lungs and provides surgeons with active localization within the bronchial tree. However, the pre-bent shape, rigidity, and size of EMN tools (around 2.7 mm diameter, Edge™ EWC Firm Tip Medial Procedure Kit, Medtronic) may have a disruptive effect on the targeting process as poor conformation of the catheter to the anatomy and associated tissue deformation impact the ability to target the tumor based on a pre-operative image of non-deformed anatomy.

EMN extends standard bronchoscopy, normally reaching the second generation of the bronchi, to the fifth generation. This equates to a 30% increase in the navigation of the bronchial anatomy.

With the aim of improving tool dexterity and anatomical reach, robotic approaches to endoscope and catheter navigation have been proposed[6]. Current commercial systems include the MONARCH (Auris Health, Inc), the Ion (Intuitive Surgical) and the Galaxy system (Noah Medical). These systems are designed to steer only the tip of the catheter, so there is no guarantee that proximal anatomy is not deformed during navigation. Additionally, relatively large diameters are retained (MONARCH 4.2 mm diameter; Ion and Galaxy 3.5 mm diameter), which limits potential anatomical reach. Alongside commercial robotic bronchoscopy systems, several research platforms have been proposed. These include a tendon-driven continuum robot[7], concentric-tube actuated steerable needles with design optimization and path planning for lung lesion targeting[8], a one-degree-of-freedom soft pneumatically driven robot for deployment from a standard bronchoscope[9,10], and anatomy-specific fully magnetic catheters for shape-forming navigation[11].

In contrast to other approaches, magnetic actuation is remote, reducing or removing the reliance on proximal mechanical and pneumatic connections and facilitating miniaturization. Indeed, tip-driven magnetic catheters have been demonstrated for cardiac and neurovascular applications at scales down to 0.4 mm[12]. However, designs that rely on axial magnetization and thus control of the tip limit the possibility for shape forming and must instead depend on the deformation of the soft structure during interaction. With a generation of appropriate magnetic fields and gradients, it is possible to control up to eight DOFs[13], making it

viable to largely improve shape-forming and deliver non-disruptive navigational capabilities of magnetic catheters beyond existing tip-driven designs. We demonstrated successful navigation of anatomy-specific fully magnetic catheters—namely magnetic tentacles[11]. This approach presents the advantages of being: (1) specific to the anatomy and guaranteed to succeed via pre-operative planning; (2) miniaturized—2.4 mm diameter; and (3) softer than the anatomy and fully-shape controllable via magnetics. These three main features have the potential to revolutionize navigation inside the anatomy. In fact, in contrast to taking advantage of functional contact, as the majority of tip-actuated catheters would, our method facilitates shaping along the length. This enables follow-the-leader motion, thus, eliminating the need for tissue interaction during introduction.

With improved non-disruptive navigation, it is also possible to move beyond biopsy and assist in unlocking the potential of targeted minimally invasive therapies, acting only on malicious cells while allowing healthy tissue and organs to continue normal function. Laser ablation is a minimally invasive technique that is used to treat a range of early cancers, such as skin, penile and esophageal[14], and can be given as either primary treatment or as adjuvant therapy. In the treatment of early-stage NSCLC, ablation techniques such as laser ablation have not outperformed stereotactic body radiation therapy, despite offering a non-ionizing technique for treatment[15]. The limited effectiveness of standard laser ablation techniques could be due to the light delivery method employed, which is through the insertion of fiber optics using needles and is indiscriminate in the heating of healthy and tumorous tissue. To help address this localization issue, the use of plasmonic gold nanoparticles that strongly absorb laser energy at a specific wavelength of light has been investigated to increase heating in tumor tissue while sparing healthy tissue[16]. Furthermore, as these particles can be molecularly targeted with tumor-specific markers, such as epidermal growth factor receptor (EGFR), they can provide further tumor localization when introduced systemically into the body. Due to the rapid attenuation of laser light in tissue, illumination sources need to be in close proximity to the tumor, which is commonly achieved through needle insertion under image guidance[17]. When performed in a minimally invasive fashion, i.e., endoscopically, this can reduce pain, discomfort, and recovery time for the patient. In this case, robotics has the potential to improve precision and safety[18,19].

In this paper, we advance the capability and application of magnetic tentacles, which we previously introduced[11], for use in targeted photothermal therapy (PTT). We present an optimization technique that can deliver a unique tentacle design capable of non-disruptive navigation to multiple targets (e.g., tumors) deep within the same anatomical structure, making multi-target therapy possible. To guarantee the target tumor is reached, we introduce supervised autonomy into our robotic platform based on real-time localization. The magnetic tentacle is synchronously inserted and magnetically manipulated to conform to the anatomy in an open-loop fashion based on pre-operative planning and path optimization. Tentacle position and shape are fed back in real-time to the clinician for monitoring the procedure.

We present a tracking method based on a magnetic localization of the bronchoscope and full-shape reconstruction of the tentacle via a Fiber Bragg Grating (FBG) sensor. Information from both sensors is combined and overlaid with the segmented pre-operative CT scan for visualization purposes. A fabrication technique is proposed to deliver integrated sensing and laser fibers into a 2.4 mm diameter tentacle with an optimized lengthwise magnetization profile.

In the present paper, we demonstrate successfully supervised autonomous navigation of magnetic tentacles in the most

complex lumina within a bronchial phantom along diverse anatomical structures in both the left and right bronchial tree. We demonstrate the possibility of deploying 1064 nm laser light at powers sufficient to cause heating and show improved targeting due to the combination of accurate navigation and therapy. Navigation demonstrations are extended to a cadaveric lung specimen, where we show an exploration of the left bronchial tree and compare the results with a standard EMN catheter. Results demonstrate the possibility for deeper navigation into the lungs while imparting less deformation on the surrounding anatomy. While the nominal diameter of the tentacle is comparable to standard tools, thus the same depth should be achieved; our experiments demonstrate an overall improvement of 37% in navigation depth.

The presented results represent a potential first step toward transforming the treatment of cancer in peripheral areas of the lungs via a more accurate, patient-specific, and minimally invasive approach.

## Results

### The magnetic tentacle bronchoscopy platform. The proposed platform for lung tumor laser therapy is represented in Fig. 1. It is comprised of two robots manipulating two external permanent magnets (EPMs), i.e., the dual external permanent magnet (dEPM) platform[11,13,20], and a standard (6 mm diameter) bronchoscope (Fig. 1a). Controlled insertion of the magnetic tentacle (2.4 mm diameter) into the primary bronchi via the bronchoscope's tool channel (Fig. 1b) is achieved using an external motorized drive system (Fig. 1c). The tip of the bronchoscope is localized using a magnetic field sensor (Fig. 1c), while the magnetic tentacle's shape is concurrently reconstructed in real-time using an FBG sensor (Fig. 1d). The two sensing modalities in combination are aligned with the anatomy and overlaid with a pre-operative CT scan to provide feedback to the clinician about the tentacle's location and shape during navigation. During the procedure, the magnetic tentacle navigates under supervised autonomy to the target tumor containing gold nanoparticles, likely administered systemically through injection during the pre-operative phase. Once in position (Fig. 1b), laser light

is delivered through the embedded laser fiber (Fig. 1d) to induce thermal ablation of the tumor.

### Phantom navigation. The magnetic tentacle was successfully navigated under conditions of supervised autonomy to four representative sub-segmental bronchi (ID 2–4 mm) in the left bronchial tree and four in the right bronchial tree, as shown in Fig. 2. These targets are problematic to explore with standard rigid tools without deforming the surrounding anatomy, which may result in more invasive navigation and mistargeting of the tumor, since targeting is performed via localization in a precomputed map of the bronchi and deformation may impact its efficacy[5].

With reference to Fig. 2, navigations e and f (Fig. 2e, f) represent two hard-to-reach locations in the left branch, and navigations b and g (Fig. 2b, g) demonstrate two cases where controlled shape-forming of the tentacle is required to follow the convoluted pathway and reach the target. Navigation a and h (Fig. 2a, h) are the least convoluted, thus mostly gradient pulling is necessary for navigation. In navigation c and d (Fig. 2c, d), we see cases where the majority of the tentacle is not bent at the base, but we can bend the tip bends to conform to the anatomy.

All navigations reported in Fig. 2 were performed autonomously within 150 s and without intraoperative imaging. Specifically, each navigation was performed according to the pre-determined optimal actuation fields and supervised in real time by intraoperative localization. Therefore, the set of complex navigations performed by the magnetic tentacle was possible without the need for exposure to radiation-based imaging. In all cases, the soft magnetic tentacle is shown to conform by design to the anatomy thanks to its low stiffness, optimal magnetization profile and full-shape control. Compared to a stiff catheter, the non-disruptive navigation achieved by the magnetic tentacle can improve the reliability of registration with pre-operative imaging to enhance both navigation and targeting. Moreover, compared to using multiple catheters with different pre-bent tips, the optimization approach used for the magnetic tentacle design determines a single magnetization profile specific to the patient's anatomy that can navigate the full range of possible pathways illustrated in Fig. 2. Supplementary Movies S1 and S2 report all

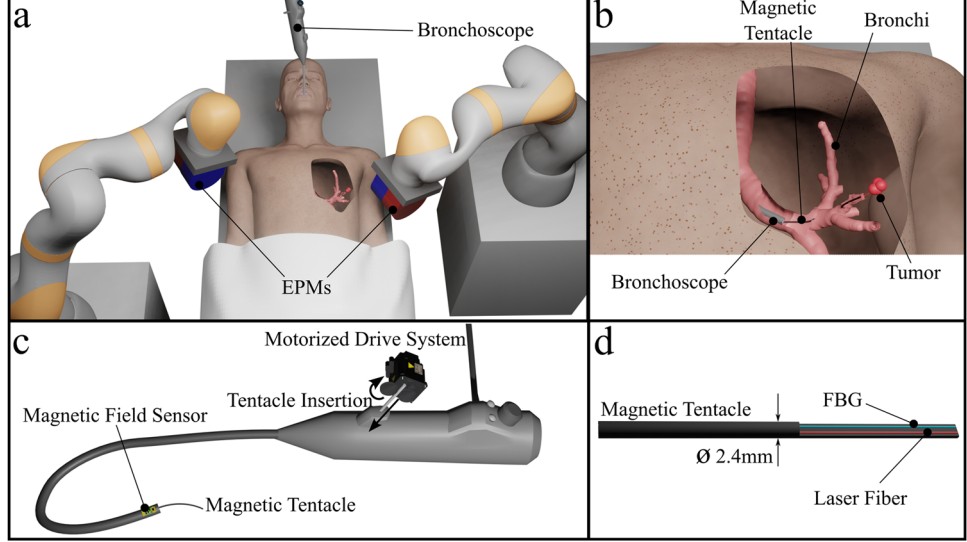

**Fig. 1 Magnetic tentacles platform description. a** Overview of magnetic tentacle delivery bronchoscope and actuation system comprised of two robotic arms, each controlling the pose of an external permanent magnet (EPM). **b** Magnetic tentacle deployment and laser delivery to a targeted tumor. **c** Illustration of the tentacle delivery system and sensing. **d** Schematic of the magnetic tentacle showing the integrated shape sensing Fiber Bragg Grating (FBG) and laser fiber.

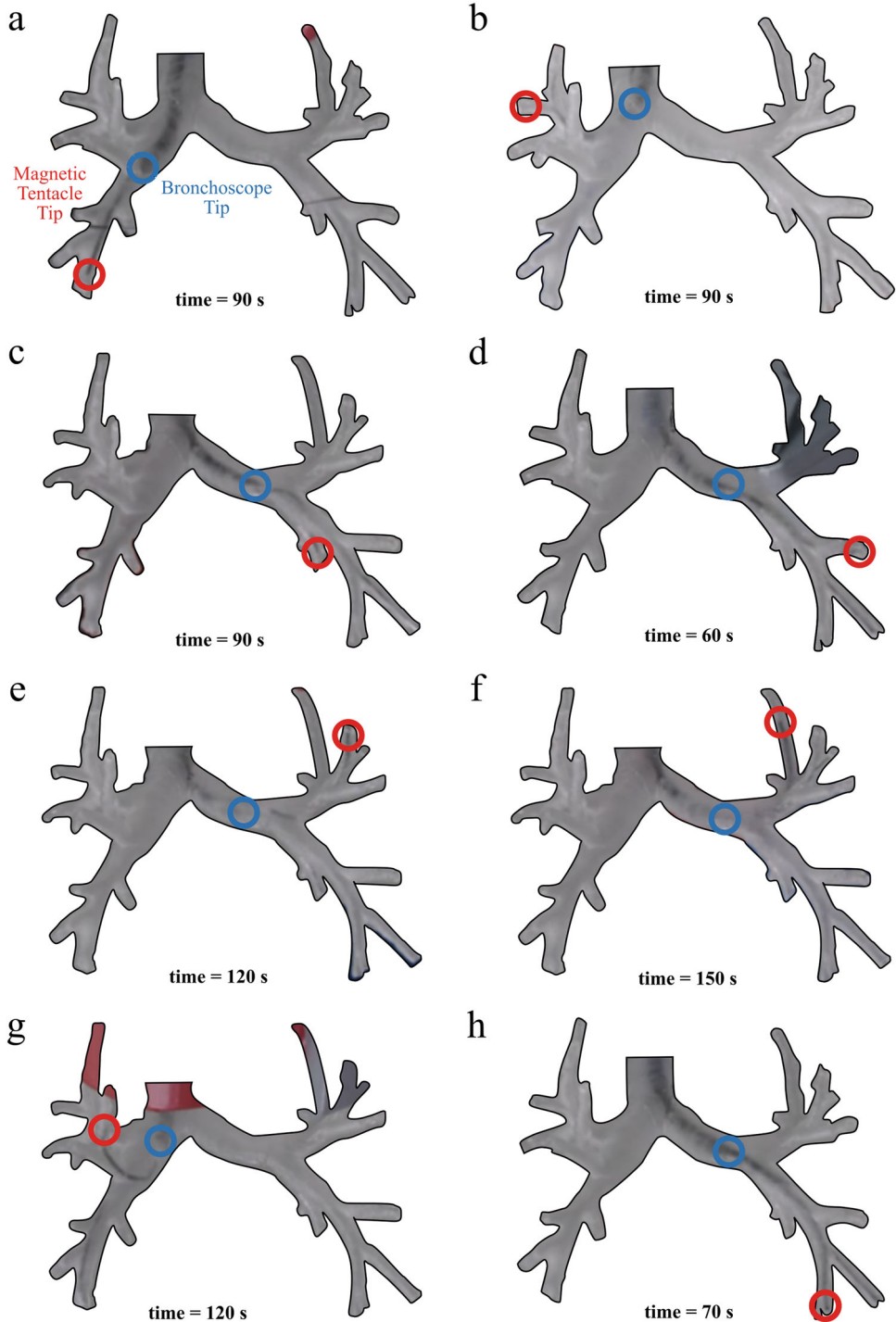

**Fig. 2 Final tentacle locations in phantom navigation experiments. a–h** Final navigation locations reached by the magnetic tentacles in eight primary targets of the sub-segmental bronchi. Locations of the bronchoscope tip (blue circle) and the magnetic tentacle tip (red circles) are indicated for each navigation, along with the associated completion time.

**Table 1 Summary of localization results for phantom experiments.**

| Scenario | Tentacle Inserted (mm) | Error (%) |
| --- | --- | --- |
| A | 70 | 9 |
| B | 57 | 2 |
| C | 133 | 15 |
| D | 71 | 11 |

the experiments. Supplementary Movie S1 shows the online tracking capabilities of the proposed platform.

In Table 1, we report the results of the localization for four different scenarios. These cases highlight diverse navigations in the left and right bronchi. The error is referred to as the percentage of tentacles outside the anatomy. This was computed by intersecting the shape of the catheter, as predicted by the FBG sensor, and the anatomical mesh grid extracted from the CT scan. The portion of the tentacle within the anatomy was measured by using "inpolyhedron" function in MATLAB. In Supplementary

Movie S1, this is highlighted in blue, while the section of the tentacle outside the anatomy is marked in red. The error in Table 1 was computed using the equation

$$err = mean_{t \in [0,T]} \frac{p_{out}(t)}{p_{in}(t)} \cdot 100$$

with $p_{in}(t)$ total number of points along the length of FBG inserted at time step $t$, $p_{out}(t)$ number of points outside the anatomy and $T$ final time.

A visualization is reported in Supplementary Fig. S1a, where we show the localization overlaid with the virtual anatomy—as used by the clinician to supervise the autonomous navigation. Supplementary Fig. S1a also reports the overall error during the full insertion process. The main sources of error are related to the possible misalignment of the angle of the bronchoscope tip and the spatial sensitivity of the FBG. In this study, one grating every centimeter was used, as a standard configuration, with a typical tip error of 1.2% of the length (from the manufacturer). We expect accuracy to improve using custom fibers with lower distance between gratings. Other sources of error are possible

consequences of misalignment between anatomy and pre-operative scan due to registration errors.

**Laser delivery**. Once the fiber was navigated to the correct location, the tissue-mimicking phantoms (with and without gold nanoparticles) were illuminated with a Continuous Wave (CW), 1064 nm diode laser. As seen in Fig. 3, for equivalent laser exposure power and duration, the phantom containing gold nanoparticles induced a peak temperature of 45 °C (Fig. 3a), while for the same phantom material without nanoparticles, a peak temperature of 33 °C was achieved (Fig. 3b), corresponding to a rise of 21.9 °C and 10.1 °C, respectively (Fig. 3c). As the initial temperature of the phantom was at approximately 20 °C a lower peak temperature was expected than biological tissue with a temperature around 37 °C. Thus, it would be expected that this higher initial temperature would cause a higher thermal dose[21,22] in more clinically relevant tissues. This, combined with the ability to navigate the fiber optic to specific locations in the lung, suggests an innovative approach for highly localized cancer treatment that would minimize damage to healthy tissue[23].

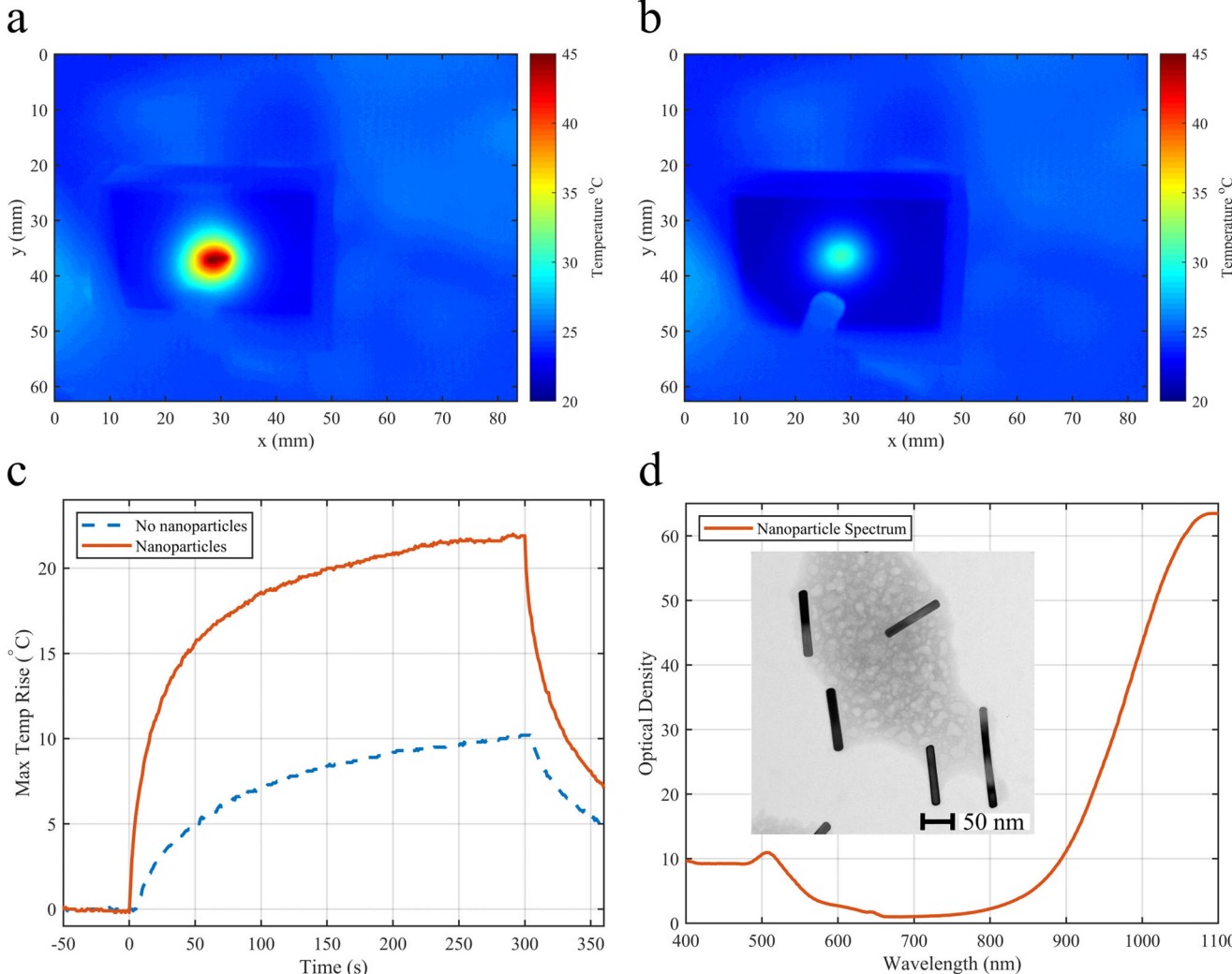

**Fig. 3 Photothermal delivery results. a** Peak temperature profile reached with targeted photothermal therapy phantom with included gold nanoparticles, *x* and *y* represent the horizontal and vertical length scales, respectively, in the plane parallel to the phantom's surface. **b** Peak temperature profile reached with targeted photothermal therapy phantom without the inclusion of gold nanoparticles, *x* and *y* represent the horizontal and vertical length scales, respectively, in the plane parallel to the phantom's surface. **c** Temperature response as a function of time for the phantom with gold nanoparticles (red line) and the phantom without gold nanoparticles (blue line); laser switched on at 0 s and switched off at 300 s, image at peak temperature point (300 s) of the phantoms shown in (**a**) and (**b**). **d** Optical response of gold nanoparticles to laser wavelength (red line).

Figure 3d shows the optical response of the gold nanoparticles. Details of the experiments are reported in Supplementary Movie S3.

**Cadaveric experiments**. Comparative navigations were performed on three main branches of the left bronchial tree of a cadaveric specimen using the magnetic tentacle and a standard bronchoscopic catheter (Edge™ EWC Firm Tip Medial Procedure Kit, Medtronic), as shown in Fig. 4. The standard catheter was operated by an expert bronchoscopist under live fluoroscopic imaging, while the magnetic navigation was performed autonomously. Details on fluoroscopic imaging configurations are reported in Supplementary Fig. S2. The branches were chosen for their diversity in terms of the amount and direction of bends along the path. The experimental procedure is presented in Supplementary Movie S4.

Navigation a (Fig. 4a) clearly presents the difference between the flexible (magnetic tentacle) and rigid pre-bent standard catheter. In fact, since the latter has a curvature that is not favorable for the specific navigation, the target could not be reached. Conversely, the magnetic tentacle can conform to the anatomy and reach the desired sub-segmental bronchi. Figure 4b shows successful navigation using both methods. Since the navigation in this example closely approximates a C-shape, the pre-bent standard tool can readily reach the segment. However, the pre-bent catheter—given the fixed radius of curvature—can only reach the upper subsegment. The third navigation (Fig. 4c) shows how both approaches can reach the target. However, the standard catheter does not conform to the natural pathway,

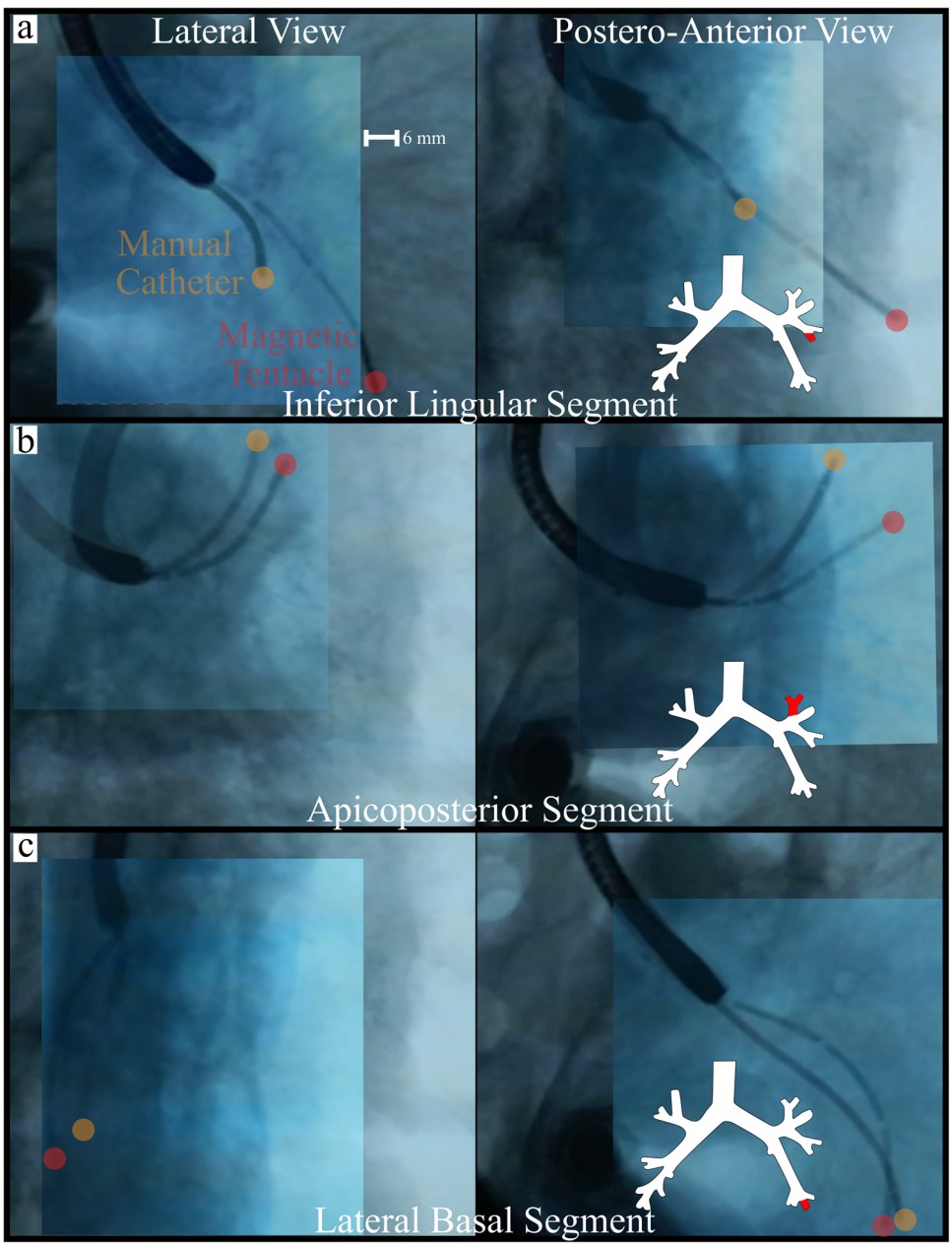

**Fig. 4 Cadaveric experiment navigation results.** Fluoroscopic images showing the final navigation locations in three primary targets in the sub-segmental bronchi of the cadaveric specimen for the manual catheter (yellow filled circle) and magnetic tentacle (red filled circle) in the lateral view (left) and posterior-anterior view (right). Regions shown are **a** inferior lingular segment, **b** apicoposterior segment, and **c** lateral basal segment. Separate fluoroscopic images from independent navigation with the manual catheter and magnetic tentacle are presented as an overlay for comparison purposes.

causing notable deformation to the anatomy. The magnetic tentacle, owing to its flexibility and full-shape control, follows the prescribed pathway without modifying the lumen. This has the potential to improve therapeutic yield since the target location would undergo less displacement with respect to its position in the pre-operative CT image.

We computed the distance between the tip of the tentacle and the catheter with respect to the tip of the bronchoscope for each navigation from the fluoroscopic images. The percentage depth was computed as the difference between the lengths normalized to the overall length. Across the three navigations performed in the cadaveric lungs, a mean improvement in navigation depth of 37% was shown with the magnetic tentacle when compared to the standard semi-rigid catheter, with the additional benefit of reduced tissue displacement.

We repeated the experiments five times for each of the targets in the same cadaveric specimen and measured the distance of the tip of the tentacle with respect to the tip of the bronchoscope (insertion point)—in lateral and postero-anterior view of fluoroscopic images (see Supplementary Fig. S3). This was used as an approximation of the navigation depth for each experiment. The maximum navigation depth in each view was used as a reference, and the difference between each repetition and the maximum navigation depth was computed. The mean error for the maximum navigation depth was found to be 0.53 ± 0.9 mm by averaging across all repetitions for both CT scan views. This demonstrates that autonomous navigation can achieve repeatable results.

In Table 2 and Supplementary Fig. S1b, we report the localization errors computed in the case of the phantom. Compared to the rigid phantom, we experience a higher localization error in the cadaveric model. This is related to the difference between pre-operative scan and navigation. Specifically, given the softness of the tissue, the anatomy undergoes changes between procedures, i.e., orientation, inflation, etc. We expect that using custom FBG sensors with reduced distance between gratings and improving the alignment of pre-operative imaging with localization (registration), these errors may be mitigated in the future.

## Discussion

In the present work, we introduce a novel approach to targeted therapy for minimally invasive lung cancer treatment. This technique has two main components: (1) a 2.4 mm diameter magnetically actuated patient-specific flexible catheter and (2) laser fiber-enabled targeted therapy. The former is based on the fabrication, localization, and full-shape control of magnetic tentacles with a specific lengthwise magnetization profile. These magnetic tentacles were optimized to the patient-specific bronchial tree and remotely actuated via collaborative control of two external permanent magnets. Laser delivery was successfully demonstrated on a tumor phantom containing gold nanoparticles. These particles can be functionalized to preferentially bind to specific tumor cells when introduced systemically and convey targeted treatment.

We present a platform capable of real-time tracking and actuation of the magnetic tentacles inside the anatomy and discuss the design and fabrication of the patient-specific tentacles. The tentacles were fabricated with an OD of 2.4 mm, which is comparable to the standard tools used in EMN. Therefore, they can enable reach to the fifth generation of the bronchi, i.e., 30% deeper than standard bronchoscopes. We demonstrated the navigational capabilities under supervised autonomy in two sets of experiments: one set in a transparent phantom and the second set in an excised specimen of human lungs. In the former, we show eight diverse navigations in both left and right bronchi—specifically, the ability to reach the sub-segmental bronchi (2–4 mm ID).

In the cadaveric lungs, we demonstrate successful navigation in three branches of the left bronchi, against the two achieved using a standard catheter, corresponding to a mean improvement in navigation depth of 37%. Moreover, we show minimal deformation of the lumen, which can maximize targeting capabilities based on pre-operative imaging.

We noticed that, while it has little influence on the navigation performance, moving the EPMs point-to-point may not always be safe. In fact, from the experiments in the phantom, we see that the robots may enter a zone reserved for the patient's body. To limit this, we will investigate safe robot planning and use larger magnets, which can allow the EPMs to be positioned further away from the patient while guaranteeing a functional magnetic field.

As part of the tissue phantom experiments, we also demonstrate laser delivery and show selective heating of tumor phantoms with and without gold nanoparticles via a magnetic tentacle-delivered laser fiber, simulating the proposed treatment paradigm and its effect on the tumor and normal tissue regions. This proposed approach has the potential to optimize minimally invasive delivery of therapy, i.e., with a soft optimal navigation technique combined with therapy targeting the tumor only. Compared to existing alternatives, this may have a fundamental impact on the quality of life of treated patients, such as removing the need for stereotactic body radiation therapy[24].

It is worth mentioning that testing the efficacy of selective heating in a cadaver model was not possible due to the impossibility of introducing plasmonic nanoparticles in a localized and controlled manner.

Open-loop control in a clinical setting could lead to some unwanted behavior during navigation and/or interactions with lung tissue due to errors in sensor readings and initial localization. Future work will focus on this while compensating for anatomical motion related to respiration and heart dynamics. We anticipate that, in this case, a high-performing closed-loop control scheme would be required. In this case, we will leverage pre-operative imaging and shape control based on FBG sensing[25]. To confirm that navigation in dynamic environments and selective tissue heating can be achieved, testing in a more realistic environment (i.e., in vivo animal model) will be pursued in future work.

Herein, we focused on a four-segment fully magnetized tentacle with maximum magnetic content along the length. However, we will investigate different distributions of the lengthwise magnetic content and optimize other mechanical parameters[26]. In addition, we will investigate an upgraded design protocol capable of deriving a globally optimal solution within the framework of these magnetic and geometric variations.

| Table 2 Summary of localization results of cadaver experiments. | | |
|---|---|---|
| Scenario | Tentacle inserted (mm) | Error (%) |
| A | 68 | 26 |
| B | 58 | 32 |
| C | 70 | 28 |

## Methods

**Patient-specific tentacles design**. From a CT scan of the full lung, we extracted a 3D model of the bronchial tree down to the sub-segmental bronchi, as shown in Fig. 5a. A point cloud of centerlines was subsequently extracted in 3D Slicer (www.slicer.org) and the proximal and distal target nodes identified. The proximal nodes represent the maximum contact-free navigation depth of the manual

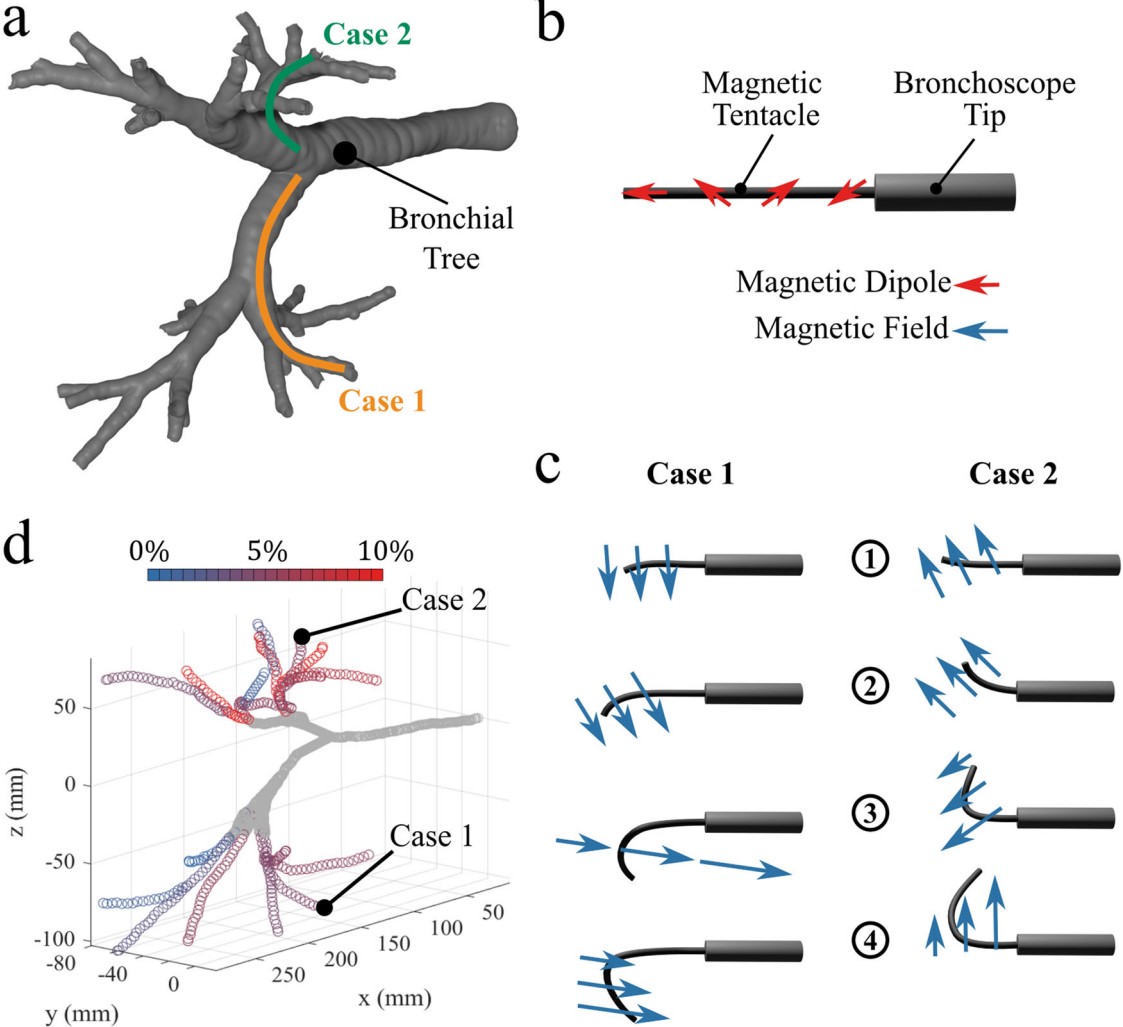

**Fig. 5 Optimal design of the magnetic tentacles.** Analysis of patient-specific design over main branches of the bronchi. **a** Evaluation of general lumina in the left (Case 1) and right (Case 2) bronchi. **b** Example tentacle magnetization profile for optimal navigation in the principal sub-segmental branches. **c** Field and field gradient actuation (blue arrows) for the main cases (1 and 2) in the four insertion steps (1)–(4). **d** The lung geometry is colored according to the percentage mean spatial error at the optimization phase; color is scaled between 0 (blue) and 10% (red) error.

bronchoscope, and distal nodes are 80 mm (total tentacle length) deeper in a diverse range of directions (eight navigations in the plastic phantom, three in the cadaveric lung). The centerline points were assigned a connectivity graph based on their distance, and the Dijkstra shortest path algorithm was used to connect the two nodes.

Our previous work introduced an algorithm to determine both magnetization directions during fabrication and applied fields and gradients during navigation[11]. This system, which we redeploy in a modified form here, represents the tentacle as a serial chain of four rigid links connected by three spherical joints. We computed the joint angles $q \in \mathbb{R}^{12}$ for which the tentacle, while inserted, would shape to the desired path—i.e., in a follow-the-leader fashion.

The virtual joints were assumed to exhibit linear stiffness derived from the elastic modulus and the respective second moment of area for the bending and twisting primitives $k$. The force of gravity and the magnetic torque and force resulting from the applied field and the spatial derivative of this field respectively combine to produce the wrench $w \in \mathbb{R}^{18}$ on each rigid link center. We minimize the static equilibrium

$$J^T(q)w - kq - g \equiv J^T(q)\begin{pmatrix} m_1 \times B \\ m_1 \cdot \nabla B \\ \vdots \\ m_4 \times B \\ m_4 \cdot \nabla B \end{pmatrix} - kq - g$$

for each insertion step; here $J(q)$ is the tentacle's Jacobian matrix, computed as a series of spherical joints, and $g$ gravitational force.

The optimization was performed for each navigation, producing a suite of eight magnetization profiles (phantom). A secondary optimization was then performed in a nested loop where the magnetization was constrained to each of the eight profiles in turn, and the desired applied fields were derived. This was looped for each of the eight navigations resulting in an $8 \times 8$ array of Euclidean norm errors between desired path (taken from the initial point cloud) and achievable navigation for every magnetization signature (Fig. 5b) and target location (Fig. 5d). Examples of response to field and gradient relative to the navigations in Fig. 5a are schematically represented in Fig. 5c.

From this result, the single magnetization signature with the lowest mean spatial error across the eight navigations (RMSE = 4.0 mm ± 1.8 mm, Fig. 5b) was selected for fabrication. Most trajectories in the bronchial tree (up to 80 mm beyond the maximum attainable depth of the bronchoscope) can be adequately approximated (RMSE = 2.5 mm ± 1.2 mm) by some planar C-shape or second-order polynomial. This diverse range of pseudo-C-shapes can then be navigated using complex transient combinations of field and spatial gradient.

**Magnetic tentacles fabrication.** Magnetic tentacles were created using a multi-step fabrication technique. To produce the tentacle's magnetic soft core, two two-part molds were designed and 3D printed (Tough PLA, Ultimaker S5, USA) to allow low-pressure injection molding, Fig. 6a. One half of the mold was designed with a cavity between the segments to allow for free flow of pre-polymer, while the other half retained a flat profile to allow the attachment of two 0.2 mm diameter nitinol (NiTi) wires precisely mid-segment along the long axis of the catheter. Additionally, an aligning cap was designed and 3D printed (Grey V4 resin, Formlabs III, USA) to ensure wires remained separated by 1 mm during the curing

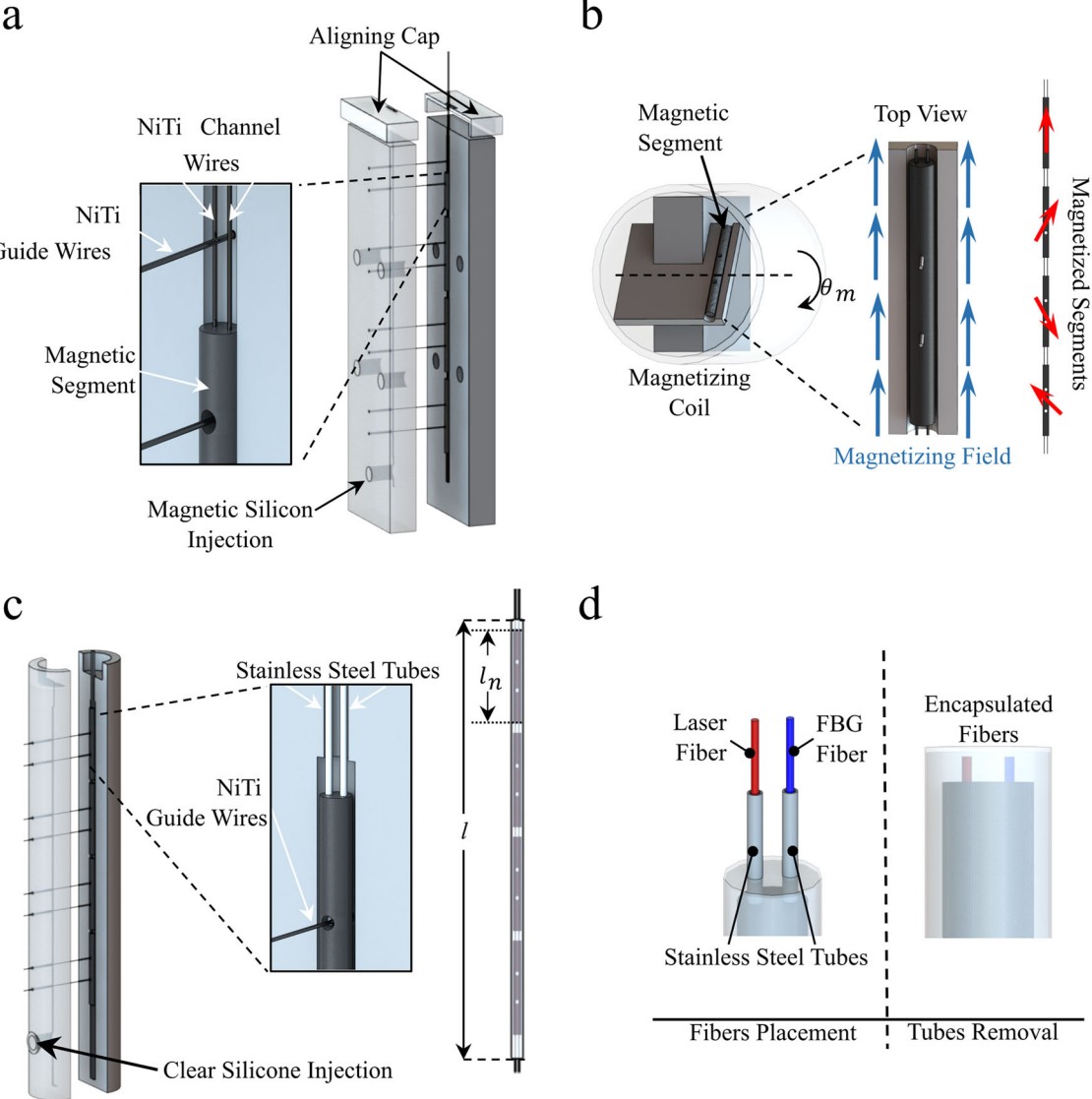

**Fig. 6 Fabrication of the magnetic tentacles.** Description of fabrication steps required to produce patient-specific magnetic tentacles. **a** Molding of the four magnetic segments with incorporated channels to house fibers. **b** Segment magnetization according to orientation prescribed by the optimization algorithm. **c** Over-molding of magnetic segments for overall tentacle bonding. **d** Placement of laser and Fiber Bragg Grating (FBG) fibers within the tentacle after over-molding.

process. Two nitinol wires (0.2 mm) were also placed into each cavity orthogonal to the long axis of the catheter (Fig. 6a) to act as an alignment feature for subsequent fabrication stages. A magnetic silicone mixture was prepared from an equal mass ratio of two-part silicone (Dragon Skin™ 10, Smooth-On, Inc., U.S.A.) combined with a 1:1 ratio by mass of hard magnetic micro-particles (Nd-FeB with an average 5 μm diameter and remanence of 0.903 T, Magnequench GmnH, Germany). The materials were mixed and degassed (ARV- 310, THINKYMIXER, Japan) for 90 s at a speed of 1400 rpm and vacuum pressure of 20.0 kPa. The degassed mixture was injected into the mold and cured at 45° for 30 min. After demolding, the structure was separated into four identical 20 mm long soft magnetic segments with a diameter of 1.8 mm.

To assign an optimized magnetic dipole moment to each of the segments, 3D-printed trays were produced for each with an associated magnetization direction (Fig. 6b). Each segment was placed into its corresponding tray using the orthogonal guide pins, ensuring precise alignment (see Fig. 6b). The sections were magnetized in a saturating field of 2.7 T[11], using an impulse magnetizer (IM-10-30, ASC Scientific, USA). Segments with assigned magnetic moments were subsequently joined in axial alignment using stainless-steel tubes (OD = 0.64 mm; ID = 0.33 mm), utilizing the axial nitinol wires as a guide before removal. The assembly of segments was then placed into a secondary mold, ensuring equal distances between the segments by aligning the guide-pins with features in the mold. The closed mold was injected with degassed silicone (Dragon Skin 10™, Smooth-On, Inc, USA) and cured at 45° for 30 min to form a clear overmold (see Fig. 6c).

To assemble the internal fibers, a 20 μm diameter FBG (4 cores, 18 sensors spaced 1 cm apart, FBGS International, Jena, Germany) and a multimode laser fiber with a core diameter of 100 ± 5 μm and a total diameter of 150 ± 10 μm (n.a. 0.22, Thorlabs Inc., NJ, USA) were first passed through a 2 mm tube and inserted through the 2.7 mm bronchoscope tool channel. The tube was inserted in the machine-driven introducer, comprising a single Bowden cable extruder actuated by a stepper motor (17HD34008-22B, Brusheng). Upon demolding of the tentacle and removal of orthogonal pins, the FBG and laser fibers at the distal end of the bronchoscope's tool channel were inserted through the stainless-steel tubes. Removal of the tubes from the molded structure resulted in a 4-segment soft tentacle with signature magnetization and two fibers running through its length, with an overall tentacle diameter of 2.4 mm and channel diameters of 0.25 mm (see Supplementary Fig. S4 for detailed microscopic image dimensional verification). The completed magnetic tentacle was bonded to the introducing tube using a thin layer of silicone adhesive (Sil-Poxy™, Smooth-On, Inc., USA). The tentacle was then retracted into the tool channel of the bronchoscope using the introducer prior to testing.

The optimized magnetic profile of the fabricated segments was visually evaluated through magneto-optical measurements (Cmos Magview-S; Matesy GmbH, Germany). Supplementary Fig. S5 represents the planar magnetization of the catheter in four configurations. The data were recorded by imaging the surface of each segment separately in four repeats, each time rotating the segments by 90°. The cross-section of each segment was also imaged from the distal and proximal ends. In both cross-sectional images, as well as horizontal images, the cavities from

the aligning pins and longitudinal channels for fibers are visible. The images were combined to present the full view of the catheter with its magnetization measured in four planes.

**Localization and shape sensing.** The soft magnetic tentacles are designed to allow for their full-shape control. This enables them to conform to curvilinear pathways while minimizing contact with tissue. However, this ultimately requires accurate localization of the point of actuation, i.e., the space around the bronchoscope's tip —considered as the workspace for the magnetic actuation.

The primary step is to find the pose of the base of the robotic arms with respect to the bronchoscope. A 4-camera optical tracking system was used for this task (OptiTrack, NaturalPoint, Inc., USA). The world frame was defined by a calibration frame fitted with optical trackers placed around the bronchoscope ("calibration rig" in Fig. 8a, c). After the bronchoscope was first inserted and manually navigated to the desired position, the calibration frame was fixed close to the trachea (see Fig. 8a, c). Additionally, the end-effectors of each robotic arm were also equipped with optical markers. Using direct kinematics, the base of each robotic arm with respect to the trachea can thus be determined. In a clinical scenario, this procedure translates to localizing the visually accessible portion of the bronchoscope outside of the patient's mouth.

To determine the pose of the distal end of the bronchoscope, we applied magnetic localization[27], which represents the starting point of the tentacle insertion within the anatomy, a digital 3D magnetic field sensor (MLX90395, Melexis, Belgium. Sensing range ±50 mT, Sensitivity 2.5 µT/LSB₁₆) was fixed at the tip of the bronchoscope. With the bronchoscope fixed in position, one of the robots was actuated, moving its EPM around the patient while measuring the norm of the magnetic field. By applying an Extended Kalman Filter, the position of the sensor was found with average errors of 5 mm across all axes. Having the position of the bronchoscope, its orientation was found by generating magnetic fields along the three inertial directions and applying a Mahony filter[28], with final errors of 3° across all axes. Lastly, the multicore FBG sensor embedded in the tentacle (see Fig. 6d) was used to accurately reconstruct its 3D shape. The FBG was calibrated with respect to the bronchoscope by deflecting the bronchoscope along its inertial directions with the magnetic tentacle inside. The overall calibration lasted approximately 5 min. Its overall error was computed by comparing the calibration's results with the data from the optical tracking system.

A live visualization system was employed to provide visual feedback to the surgeon regarding the shape and position of the magnetic tentacle in the anatomy when not using fluoroscopy. A 3D representation of a segmented pre-operative CT scan was loaded into this virtual environment along with the current shape and position of the magnetic tentacle. The start position of the tip of the magnetic tentacle with respect to the anatomy was at the tip of the bronchoscope. Using the insertion speed of the magnetic tentacle, along with the shape of the FBG, a real-time, virtual representation of the shape and position of the magnetic tentacle was constructed. This visualization tool allowed the surgeon to have knowledge of the tentacle's position and shape without introducing radiation-based imaging. A demonstration of the use of the visualization system can be seen in Supplementary Movie S1.

**Magnetic actuation.** The magnetic tentacles were actuated using the dEPM platform[11,13,20], which is comprised of two independent permanent magnets, able to generate space- and time-varying controlled magnetic fields. Specifically, we aim to use the overall magnetic field direction to shape the tentacle to the anatomy and the additional gradient created by the relative motion of the EPMs ($\delta$) to pull the tentacle. In principle, the magnetized sections of the tentacle align with the overall field and the gradient creates an attraction/repulsion force that pulls the catheter through the anatomy (Fig. 7). Notice that, compared to using only one EPM, we can produce both attraction and repulsion; also, the strength of field and gradients can be independently controlled.

The optimization routine described in Fig. 5 produces a desired field $B$ and desired gradient $\nabla B := \partial B / \partial v$, as shown in Fig. 7. The direction $v$ is selected based on the direction of pulling force, i.e., tangent to the anatomy's centerline. We employ the dipole model to control the EPMs to the required pose and guarantee control of the desired field and gradient. According to the model, the field and gradient are

$$\begin{cases} B = \frac{\mu_0}{4\pi|\rho+\delta|^3}\left(3vv^T - Id\right)\mu + \frac{\mu_0}{4\pi|\rho-\delta|^3}\left(3vv^T - Id\right)\mu \\ \nabla B = \frac{3\mu_0}{4\pi|\rho+\delta|^4}\left(Id - 3vv^T\right)\mu + \frac{3\mu_0}{4\pi|\rho-\delta|^4}\left(Id - 3vv^T\right)\mu \end{cases}$$

Here we force the magnetic dipole of the EPMs to be parallel $\mu_1 = \mu_2 = \mu$; $Id \in \mathbb{R}^{3\times3}$ identity matrix and $\rho$ is the distance between each EPM and the center of the workspace. The EPMs are controlled to align their magnetic dipole to the desired field direction $\hat{B}$, according to the dipole model, by solving

$$\hat{\mu} = |3ve^T - Id|\left(3vv^T - Id\right)^{-1}\hat{B}$$

Notice that the field intensity cannot be controlled by changing $|\mu|$, which is constant for permanent magnets. To reproduce the desired field and gradient strength required by the optimization routine, we regulate the position of each magnet with respect to the tentacle $p_i = (\rho \pm \delta)$ by solving the set of equations

$$\begin{cases} \frac{1}{|\rho+\delta|^3} + \frac{1}{|\rho-\delta|^3} = |\left(3vv^T - Id\right)^{-1}|\frac{4\pi|B|}{\mu_0|\mu|} \\ \frac{1}{|\rho+\delta|^4} + \frac{1}{|\rho-\delta|^4} = |\left(Id - 3vv^T\right)^{-1}|\frac{4\pi|\nabla B|}{3\mu_0|\mu|} \end{cases}$$

with respect to $\rho$ and $\delta$. Notice that matrix $\left(Id - 3vv^T\right) = -\left(3vv^T - Id\right)$ is always invertible.

The combination of field alignment—i.e., tentacle magnetization alignment to the overall field (configuration 1 in Fig. 7)—and gradient pulling (configuration 2 in Fig. 7) guarantees smooth navigation of the tentacle within the anatomy. This is ensured when the applied field and gradients are controlled according to the pre-operative optimization routine. Effective autonomous navigation was supervised via the proposed localization approach.

**Phantom setup.** To demonstrate the capabilities of the patient-specific magnetic tentacles, we performed a set of experiments in a clear phantom of the lungs (Fig. 8a), 3D printed (Clear V4 resin, Formlabs III, USA) from real patient's CT scan (Lung Image Database Consortium image collection

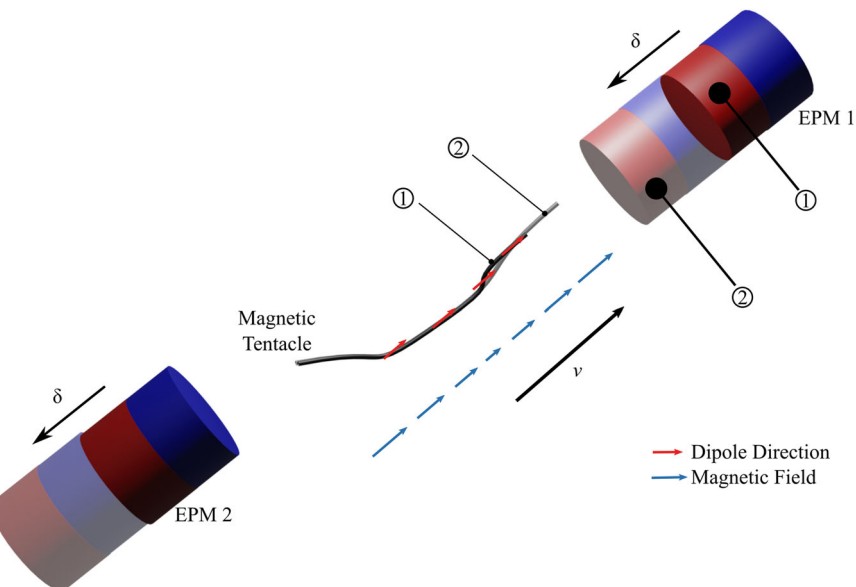

**Fig. 7 Magnetic tentacle actuation principles.** Description of magnetic steering based on field-magnetization alignment (1) combined with tip-dragging in a tangential direction to the anatomy's centerline ($v$) realized via gradient pulling and relative translation ($\delta$) of the external permanent magnets (EPMs) (2).

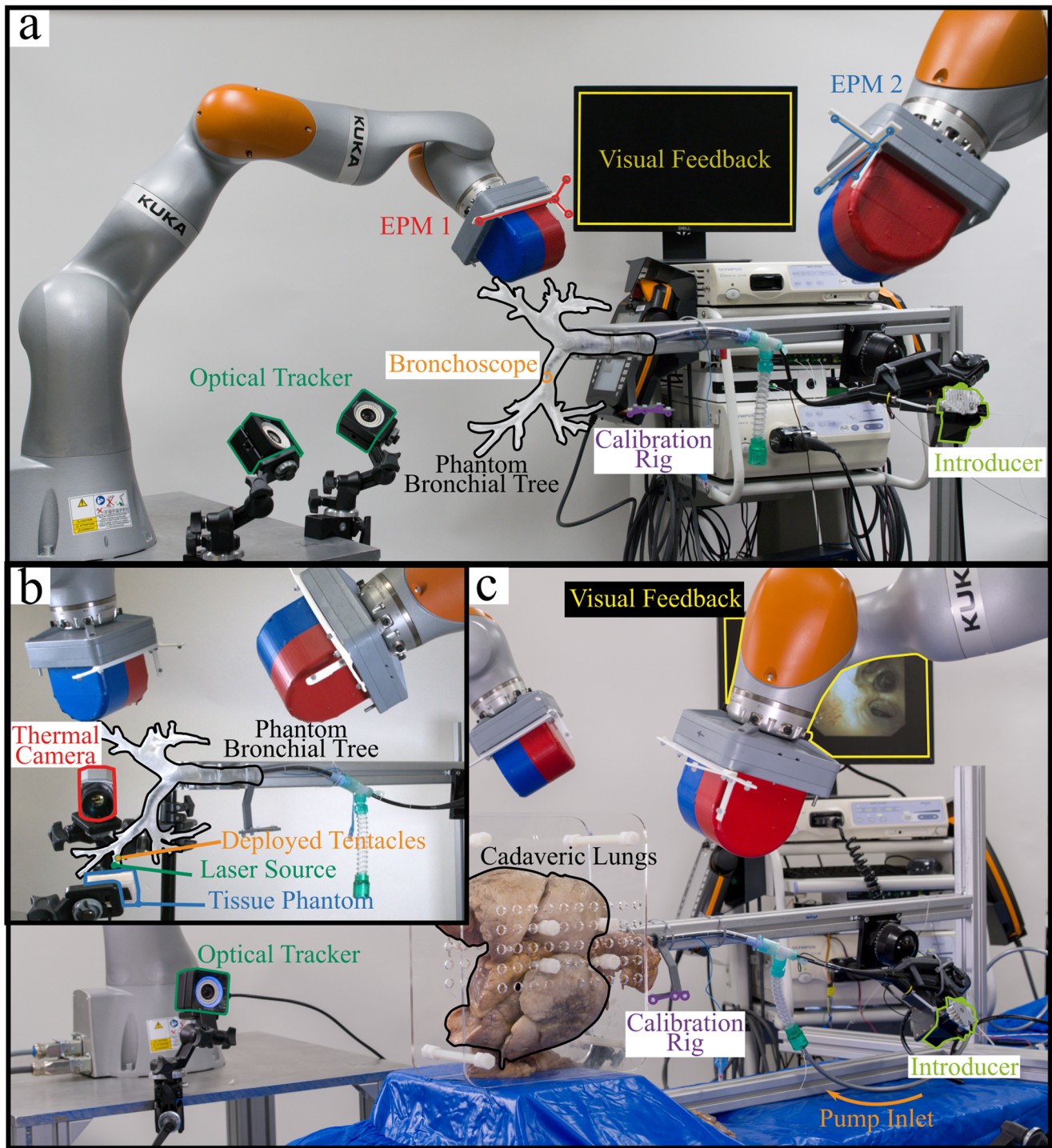

**Fig. 8 Experimental setup. a** Phantom experiments setup. **b** Photothermal energy delivery experiments setup. **c** Cadaver experiments setup.

LIDC-IDRI-0807—www.cancerimagingarchive.net). This guarantees visual tracking of the tentacle to demonstrate its navigational capabilities.

The experimental setup, reported in Fig. 8, was composed of the dEPM platform—2 robotically actuated EPMs, a 4-camera optical tracking system (OptiTrack, NaturalPoint, Inc., USA, with sub-millimeter accuracy), a motorized introducer and visualization screen. The dEPM platform was employed for field actuation, synchronized with the introducer, and commanded by a pre-operative optimal field.

The magnetic tentacle was deployed through the tool channel of the bronchoscope (BF-1T180; Olympus Corporation) and inserted by the introducer from the proximal section of the bronchoscope. The bronchoscope was manually navigated to the primary bronchi (right and left), localized, and used to deploy the robotically actuated tentacle. From the primary bronchi, the aim of the magnetically manipulated tentacle was to target deep anatomical structures which

are challenging or impossible to reach using standard tools and other robotic platforms due to their larger relative diameter and/or stiffness.

The EPMs were constrained to respect a center-to-center distance of 50 cm to avoid attraction and collision with the phantom.

**Laser experiments setup**. A 1064 nm CW laser diode with integrated multimode optical fiber (n.a. 0.29) and peak optical power output of 2 W was used as the illumination source for the PTT study (B2-A64-2000-15C, Sheaumann Laser In, MA USA). The laser was situated in a 14-pin butterfly laser diode mount (LM14S2, Thorlabs Inc, NJ, USA), which was connected to a thermoelectric cooler (TEC) controller (TED200C, Thorlabs Inc, NJ, USA) and a 4 A laser diode driver (LDC240C, Thorlabs Inc, NJ, USA). The output of the optical fiber built into the diode was coupled into the separate fiber embedded in the tentacle using an SMA

to SMA mating sleeve (ADASMA, Thorlabs Inc, NJ, USA). To account for the optical power loss from this approach, the peak power was measured at a distance of 20 mm from the output of the complete illumination system using a power meter (S370C, Thorlabs Inc, NJ, USA). The laser diode driver was adjusted to give a peak optical power output of $0.5 \pm 0.05$ W onto the tissue-mimicking phantom, which is much lower than powers typically used in clinical laser ablation[29]. Tissue phantoms were manufactured using an adapted protocol from Rickey et al. (1995)[30]. Then, 3.6% (by mass) of agar powder (Acros Organics, Geel, Belgium) was added to deionized, degassed, and filtered water (82.9%, by volume) and polydisperse (4–45 μm) glass beads (5%, by weight) (Honite 22, Guyson, North Yorkshire, UK). The mixture was heated to approximately 90 °C and maintained at this temperature for 30 min while continuously degassed using a magnetic stirrer and hot plate. The mixture was left to cool to 70 °C before adding in (1%, by mass) a preservative, Germal$^+$ (Gracefruit, Stirlingshire, UK) and glycerin (10%, by mass) (Merck Life Science, Dorset, UK) and mixed. Once fully mixed, half of the mixture was poured into a rectangular plastic container ($40 \times 25 \times 10$ mm), while gold nanorods (D12-10-1064-NC-PBS-50-1, Nanopartz Inc, CO, USA) were added to the remaining solution ($2.5 \times 10^{10}$ nr/ml) allowed to mix, then this solution was poured into a separate container, and allowed to set. Once these phantoms had set, they were stored in the laboratory fridge at 20 °C until needed.

A thermal camera (thermoIMAGER TIM 640, Micro-epsilon-Messtechnik GmbH & Co KG, Ortenburg, Germany) was focused on the surface of the phantom, which was positioned 20 mm from the exit of the bronchi (coincident with the laser fiber output), as shown in Fig. 8b. It was used to record the surface temperature of each phantom before, during and after illumination by the CW laser. Total exposure time was 5 min with a frame record every 1 s. Post-processing was done using MATLAB 2019a (Mathworks Inc, MA, USA). The laser-phantom distance was selected to facilitate thermal measurements using an external thermal camera. In the clinical scenario, we expect the target to be closer to the laser source, as well as not in front of it. The former would improve thermal ablation due to higher absorption. Steering optics can be adopted to guarantee exposure of areas that are not directly in front of the catheter.

The fiber used in this study was selected with a bending radius of approximately 15 mm, less than the bending of the tentacle in any of the navigations. Therefore, bending losses were minimized. We also computed the overall loss along the length of the fiber for laser delivery as 22.5 mW, which results in a negligible heat increase in the fiber cladding/coating. This guarantees no notable signal loss or damage to the FBG.

**Cadaveric setup**. We performed a set of three navigations in a Thiel-embalmed cadaveric specimen of the bronchi. The setup, represented in Fig. 8c, is composed of the dEPM platform used for the actuation of the magnetic tentacle, a monitor for visual feedback from the bronchoscope's camera and localization, a motorized introducer, and a calibration system. The latter is formed of a set of a 4-camera optical tracking system and a calibration rig. The rig is placed at the beginning of the trachea and represents the real case scenario, i.e., calibration with respect to the patient's mouth. This system is used for real-time localization of the magnetic tentacle in the bronchi. The cadaveric lungs and accompanying organs were positioned as if turned onto their left side and supported within an acrylic frame.

The bronchoscope was inserted through a standard cannula (size 9 ET tube, Seal® Cuff Tracheal Tubes, Smiths Medical), secured at the level of the trachea by means of an inflated cuff and navigated visually to the primary bronchi. In this procedure, visual feedback from the bronchoscope's camera was used. Subsequently, autonomous navigation was enabled. A DC compressor pump (DP0102, Nitto Kohki, Japan) was connected to the cannula to provide fixed inflation of the cadaveric lungs during the procedure. The procedure was performed by an expert bronchoscopist.

To visualize the advancement of the magnetic tentacle through the bronchi, fluoroscopic images (Siemens Siremobile, Siemens, Germany) were intermittently captured. Details of the imaging setup can be found in Supplementary Fig. S2. Two projections were achieved, as is the usual protocol during most radiographic procedures. Demonstrating a 3D object on 2D images makes an orthogonal view essential within radiography[31]. It was particularly useful while imaging the cadaver to aid the accurate localization of the bronchoscope and internal equipment within each selected pathway and bronchi. In virtual bronchoscopic navigations within clinical settings, X-ray guidance has been necessary for accuracy and real-time ability to prove location[32]. However, in our experimental protocol, imaging was only used for validation purposes, while supervised autonomy only relied on pre-operative optimization and online localization.

To achieve the lateral view of the lungs, the Image Intensifier (II) (Supplementary Fig. S2a) was moved over the lungs in its neutral position and set above the setup with the X-ray tube underneath the table. For the lateral projection, the exposure factors manually set were 61 kV and 1.5 mA. For the postero-anterior (PA, Supplementary Fig. S2b) projection, the C-arm II was turned from its neutral position 90° clockwise, whereby the X-ray tube was at the posterior aspect of the cadaver and the II was at the anterior aspect of the cadaver. The exposure factors manually selected for the projection were lower than that of the lateral to account for the reduction in the amount of tissue there was to penetrate. The factors were 53 kV and 0.5 mA.

The equipment utilized within the setup around the lungs, such as the frame, table and other supporting objects, were radiolucent. This limited artifacts on the radiographs and created a clear view of the required anatomy. We imposed a center-to-center distance between the EPMs of 50 cm to guarantee no cross-attraction and collision with the frame containing the lungs.

**Reporting summary**. Further information on research design is available in the Nature Portfolio Reporting Summary linked to this article.

## Data availability

All data are available in the main text or the supplementary materials.

## Code availability

The code that supports the findings of this study is available, but restrictions apply to the availability of some sections, which were used under licence, and so are not publicly available. Code generated by the authors is available from the authors upon reasonable request, and an application for licence for protected components can be made to FBGS.

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

## Acknowledgements

The authors express their gratitude to the donor who so generously donated their body to the University of Leeds for others to learn from. We also thank Sarah Wilson and Charlotte Coleman for their help in setting up and running the cadaveric experiments reported in the paper. J.M. acknowledges support from the Hollins Scholarship. Research reported in this article was supported by the Engineering and Physical Sciences Research Council (EPSRC) under grants number EP/R045291/1, EP/P027687/1 and EP/V009818/1, and by the European Research Council (ERC) under the European Union's Horizon 2020 research and innovation programme (grant agreement No. 818045). Any opinions, findings and conclusions, or recommendations expressed in this article are those of the authors and do not necessarily reflect the views of the EPSRC or the ERC.

## Author contributions

Conceptualization: G.P., J.C., T.d.V., Z.K., M.B., P.L., P.V. Methodology: G.P., J.C., T.d.V., Z.K., M.B., P.L., R.H., P.V., J.M. Investigation: G.P., J.C., T.d.V., Z.K., M.B., P.L., C.P., J.M., K.B., C.P., R.H., P.V. Visualization: G.P., M.B. Funding acquisition: J.M., R.H., P.V. Project administration: G.P., J.C., P.V. Supervision: G.P., J.C., P.V. Writing—original draft: G.P., J.C., T.d.V., Z.K., M.B., P.L. Writing—review & editing: G.P., J.C., T.d.V., Z.K., M.B., P.L., J.M., R.H., P.V.

## Competing interests

The authors declare no competing interests.
