## [Peer Review File · Communications Engineering]

Reviewers' comments:

Reviewer #1 (Remarks to the Author):

In this manuscript, the authors design a new surgical instrument integrating the magnetic tentacles (they previously published) with the FBG sensors for shape sensing and an optical fibre for targeted laser therapy. They demonstrate this multifunctional instrument in a phantom and a cadaver. Clinical unmet needs and limitations of current approaches are well defined. The benefits of the new device/approach are demonstrated and compared with existing instruments. The manuscript is well-written, with the necessary details presented. The references cover the relevant work published.

Please find my comments below:

- "This approach can potentially save patients from one of the most life-threatening cancers" is a strong statement. It potentially takes much more than a device to save patients, so I suggest changing this statement.
- "The error is referred to as the percentage of tentacles outside the anatomy" this error needs further explanation; it is difficult to understand from the description. Do the tables report the error in the final position? This error changes with time and reaches very high values. Is this a concern in surgery? What is considered high or low, and why? Fig. S1 needs more explanation to clarify these points.
- "Average error in penetration depth" in the abstract is not clear what it is (is it an error in repeatability or error in position?) . "Penetration depth" is usually used for energy (often referring to EM wave, ultrasound etc.) entering a material, so the reader may think it is the laser penetration depth. To avoid confusion, authors might want to continue using "navigation depth". In the text, it is not clear what is averaged to get 0.53 mm. Is this an error? Is this the error that was mentioned in the abstract?
- "... using an external motorised drive system (Fig. 1B)" readers may expect to see the motorised drive system in Fig. 1B.
- Can the authors please comment on the safety of the temperature increase in a human body; they mention it will be higher than that of the phantom, but no safety discussion around this is presented.
- More details need to be provided regarding the laser exposure. It is not clear what the distance from the fibre output end to the tissue-mimicking phantom is. Is this distance reasonable in surgery? This approach assumes the tumour is in front of the fibre. Can the laser energy be directed to a tumour on the side of the bronchi? These points need to be clarified to ensure the feasibility of this approach.
- Please check the format of the references

Although existing magnetic catheter-based robotic technologies show great flexibility and have been demonstrated in different clinical applications, this paper is comprehensive about the use of magnetic tentacles integrated with shape sensing and therapy components for the treatment of bronchial tumours. The work will be interesting to the engineers in the medical robotics field and to the general readership of Communications Engineering.

Reviewer #2 (Remarks to the Author):

This is a very interesting paper that demonstrates the magnetic actuation of a tentacle to deliver laser ablation in multiple locations of the lung. Several major claims are made including the ability of the tentacle to navigate autonomously and without deformation of the surrounding tissue via feedback from Fiber Bragg Grating sensors. Optimization of the magnetic actuation methods allow for more than

one pathway so a single tentacle can address multiple target locations. The clinical goal is to treat the target cells using laser ablation techniques. This paper will be of great interest to the community because of its contributions to soft, medical robotics. The introduction of autonomous magnetic navigation to multiple target areas using 4 degrees of freedom and the integration of therapeutic tools into a miniaturized robot are contributions with merit. This paper will help advance the field and influence the technological capabilities of soft robots in minimally invasive surgical applications. The methods used by the author result in a novel contribution to the field. However, further evidence is required to strengthen the conclusions of this work as detailed in the questions below. In general, the use and safety of lasers in bronchoscopy may be further explained in the Introduction. The authors discuss only a few state of the art robots for bronchoscopy and magnetically actuated robots. To frame the work more effectively in the field, it is recommended that the authors expand on the review of relevant and recent works. A further review of laser surgery, particularly with robotics, soft magnetic robots, and other robots developed for bronchoscopic procedures would strengthen the novelty claimed in the paper. Some starting points for references on state-of-the-art robots in bronchoscopy and the use of lasers in surgery are:

1. York, P. A., Peña, R., Kent, D., Wood, R. J. (2021). Microrobotic laser steering for minimally invasive surgery. *Science Robotics*, 6(50), 5476.
2. Lee, H.C., Pacheco, N.E., Fichera, L., Russo, S. (2022). When the End Effector Is a Laser: A Review of Robotics in Laser Surgery. *Advanced Intelligent Systems*, 4(10), p.2200130.
3. Swaney, P. J., Mahoney, A. W., Hartley, B. I., Ramirez, A. A., Lamers, E., Feins, R. H., Alterovitz, R., Webster, R. J. (2017). Toward Transoral Peripheral Lung Access: Combining Continuum Robots and Steerable Needles. *Journal of Medical Robotics Research*, 2(1).
4. Van Lewen, D., Janke, T., Lee, H., Austin, R., Billatos, E., Russo, S. (2023). A Millimeter-Scale Soft Robot for Tissue Biopsy Procedures. *Advanced Intelligent Systems*, 2200326.
5. Wang, L., Zheng, D., Harker, P., Patel, A. B., Guo, C. F., Zhao, X. (2021). Evolutionary design of magnetic soft continuum robots. *Proceedings of the National Academy of Sciences of the United States of America*, 118(21), e2021922118.

One of the major claims of the paper is the ability of the tentacle to conform its shape at multiple points along its length so that the tentacle is non-disruptive to the lung walls. However, it seems that there are instances when the sensor detects that the tentacle is outside of the lung which implies that it is contacting the lung walls. In the phantom navigation experiments, the tentacle looks as if it is contacting the lung to make turns (Navigation B, Movie S1). The potential deformations due to this contact should be quantified to strengthen the claim of fullshape control. The methodology employed for minimizing contact should also be explained as this is unclear. Contact minimization is claimed in line 450 and both Cadaver and Phantom Experiments in the paper but not explained in detail. Similarly, the localization error was stated to be 5 mm which is significant at smaller lung branches in the periphery. Does this localization error affect shape control? Magnetic actuation has the significant benefit of controlling multiple degrees of freedom simultaneously with the same set of EPMs. How does the design of the tentacle such as spacing, length, and amount magnetic segments affect its ability to conform to pathways? Further comments on these design choices would be helpful. The authors state that optimization was performed for each of 8 navigations to produce the optimal magnetization profiles for

each navigation. After this, the 8 different fields to achieve each navigation was derived for each of the 8 magnetization profiles. The optimization takes the magnetization profile with the minimum error. Is it possible there is a more optimal magnetization profile that is not considered as a result of this process? Can the magnetization profile be optimized once to all 8 navigations rather than just each individual navigation? This question pertains to the experimental setups and the strength of the EPMs. What about the case when there is additional anatomy between the lungs and the EPMs as is the case in a real clinical setting? If the distance from the EPMs to the tentacle required for navigation is less than is allowed due to the anatomy, will there be any method to account for this? The authors state that the location of the tip relative to the tip of the bronchoscope is determined based on the speed at which the tentacle is advanced by a stepper motor. However, it is also mentioned that there is a pulling force imposed on the magnetic segments by the EPMs. How does the pulling force effect the speed and, consequently, the location of the tip? Potential contact interactions with the lung walls should also be taken into account as this can affect the speed and direction of the tentacle. Laser ablation is a therapeutic technique that has significant potential in treating these types of lung cancers. There are a few questions on how it is used in this paper. The authors claim that the laser is highly localized. However, the area affected by the laser seems large in Fig. 3 relative to the scale of the lung anatomy. Can the tentacle navigation be used to control the size of the affected area? Are there any unexpected losses in laser power as a result of the fiber being bent in the anatomy? Will the laser cause heating of the tentacle and affect the sensitivity of the magnets or the shape sensing FBG? In a clinical situation, the target cells will likely not be directly in front of the robot but rather along the walls of the lungs. The laser ablation was only demonstrated directly in front of the tentacle. Is the tentacle able to steer the laser so that it can aim towards the lung walls and achieve a close enough distance to effectively ablate the cells? Further, can this maneuver be optimized so that the robot minimizes contact with the lung walls as is claimed? In the Cadaveric Experiments, analysis of the repeatability is a difference between maximum penetration in each view of the fluoroscopic images. It is unclear whether the mean value reported is in a specific view or across all views in the fluoroscopic images. Further, reporting a standard deviation for this analysis would strengthen the claim of repeatability and prove that the mean value can be reliably repeated. Does the approximation error of the second order polynomial compound with any sensor error that is attributed to limited resolution? The resolution of the FBG sensor is claimed to be a source of error. However, its accuracy is not analyzed. What is the error of the FBG sensor? More details on the calibration method would likely help with this explanation as well.

Reviewer #3 (Remarks to the Author):

The authors present a system for targeted photothermal cancer therapy in the respiratory system using a flexible magnetic guide wire containing a laser fiber for local heating and a FBG fiber for shape sensing. The “magnetic tentacle” is steered into several lung branches using two external permanent magnets. The aim of this study is clear and the shown experiments demonstrate the capabilities of this method well. The paper is well-written. The validation of the method in a cadaver lung is good. However, the referee feel that several major points should still be clarified before publishing. My detailed comments are below.

Major comments:

1. The experiment for Fig. 2 (Movie S1) is unrealistic as the physical boundaries of a human torso are not

respected. The two external magnets move freely in space and do not appear to have constraints in their movement. Realistic orientations and distances of the external magnets in relation to the bronchial tree should be considered.

2. The localization error (Table 1 and 2) is not well defined. It is not clear to the referee how the spatial error is measured. Is it an average over the whole tentacle? Do the authors take the length of the tentacle into account to normalize the error? A clear mathematical definition is necessary.

3. Fig. 3 illustrates the photothermal delivery results on phantom. It is very confusing. Does the material mimic tissue property for laser absorption? Where do the gold nanoparticles come from, and for what purpose? Is the result relevant to the tentacle navigation and localization? If the authors believe it is highly relevant, this should be shown in conjunction with the segmental bronchi phantom.

4. The overall flow of the manuscript needs careful consideration and modification. The referee believes it is easier for the readers to follow if the tentacles structure, fabrication, and actuation (Fig. 5-7) are discussed earlier, before the navigation results (Fig. 2 and Fig. 4).

5. Why is an extended Kalman filter applied to determine the position of the bronchoscope? How was the error determined? How long did the calibration of the position and orientation take?

6. What is the physical accuracy of the new localization method? Can the authors give quantitative evaluation about this on a regular spatial grid as a ground truth? How much does the localization error depend on the magnetic localization or the tip of the bronchoscope?

7. Does the permanent magnetic actuation affect the magnetic localization? How was this problem solved?

Additional comments:

1. The figures have inconsistent quality with respect to readability, font size, dimensions and overlays.

2. The abstract is too long and too detailed. Main novelties should be summarized concisely.

3. Fig. 2:

a. The image quality is quite low.

b. Why do the authors not show the whole image and mark the relevant area separately?

c. What is the artifact in D on the right side?

4. Movie S1: It would be beneficial to pause the video for a few seconds when the target is reached.

5. Fig. 5: The figure labels do not read left-to-right and up-to-down.

6. What is the grade/remanence field of the magnetic particles?

7. Spelling mistake: Mahony filter

8. Reference to the Mahony filter is missing

Revision of COMMS-ENG-23-0016-T

We thank the editors and reviewers for their constructive feedback on the paper's form and content. We gladly applied all changes requested and responded to all the comments, as follows in the present document.

In the following, our response was highlighted with **red text**. In the revised manuscript:

- **Removed text** indicates major reduction in the text.
- **Red text** indicates additional language, in response to reviewers' comments.
- **Blue text** indicates text of the original manuscript which we highlighted in response to a specific comment from the reviewers.
- **[Ri-j]** labels where the reviewers can find changes reflecting the comment **j** of the reviewer **i**.

Reviewer #1

In this manuscript, the authors design a new surgical instrument integrating the magnetic tentacles (they previously published) with the FBG sensors for shape sensing and an optical fibre for targeted laser therapy. They demonstrate this multifunctional instrument in a phantom and a cadaver. Clinical unmet needs and limitations of current approaches are well defined. The benefits of the new device/approach are demonstrated and compared with existing instruments. The manuscript is well-written, with the necessary details presented. The references cover the relevant work published.

We thank the reviewer for their review, and we are glad they find the paper well-written and presented. Please, find our response in the following and appropriate changes in the text.

Please find my comments below:

- "This approach can potentially save patients from one of the most life-threatening cancers" is a strong statement. It potentially takes much more than a device to save patients, so I suggest changing this statement.

We modified the abstract as response to another reviewer's request and changed the statement. See label **[R1-1]**.

- "The error is referred to as the percentage of tentacles outside the anatomy" this error needs further explanation; it is difficult to understand from the description. Do the tables report the error in the final position? This error changes with time and reaches very high values. Is this a concern in surgery? What is considered high or low, and why? Fig. S1 needs more explanation to clarify these points.

We thank the reviewer for these comments and acknowledge that the results in Table 1 may be confusing without further explanation.

The error presented in Tables 1 and 2 represents the percentage of tentacle within the 3D mesh of the anatomy at every time step, averaged over the total insertion time. This can be expressed by the equation:

$$err = \text{mean}_{t \in [0, T]} \frac{p_{in}(t)}{p_{out}(t)} \cdot 100$$

with $p_{in}(t)$ number of points along the length of FBG inside the anatomy, $p_{out}(t)$ number of points outside the anatomy and T final time.

The change in error % throughout the experiment happens for two main reasons:

For the phantom experiments this error is mainly attributed to the fact that we assume that the amount of tentacle inserted is perfectly represented by the introducer. This does not account for slipping of the introducer gears and buckling of the tentacle. This results in periods of high error which typically return to a lower value after the tentacle experiences a sudden movement in the desired direction (can be seen in Video S1 at min 1.05). This can be seen in the error graphs for the phantom experiments.

For the cadaver experiments, along with the introducer issues mentioned above, we also assume that the lung structure of the cadaveric specimen is identical to the pre-operative scan. Although the pre-operative scan gives a good indication of the shape of the path we intend to navigate, there can be slight differences in pathway shape, especially in the distal ends of the bronchi. This can be seen by the error increasing as the navigation time increases. The main reason behind this discrepancy between scan and actual pathway shape, is due to a change in setup between scanning of the specimen (laying on the coronal plane), and experimental setup (in the sagittal plane). This change in setup was necessary to fit the cadaveric specimen in CT scanner.

Registration, i.e., alignment of pre- and intra-operative imaging can have a strong impact in navigation, and it is one important source of error in the current practice. This, together with segmentation errors can cause errors in localization. The role of this paper was to solve the challenges of navigating in a partially unknown anatomy using magnetic navigation and localization. This was successfully proven, but we agree that improving registration and segmentation can guarantee further enhancement the platform's capabilities. In the future, we will focus our attention on improving pre- and intra-operative imaging, which will definitely pose challenges, opportunities and important findings.

We added these considerations to the revised version of the paper and labelled them with [R1-2].

- "Average error in penetration depth" in the abstract is not clear what it is (is it an error in repeatability or error in position?). "Penetration depth" is usually used for energy (often referring to EM wave, ultrasound etc.) entering a material, so the reader may think it is the laser penetration depth. To avoid confusion, authors might want to continue using "navigation depth". In the text, it is not clear what is averaged to get 0.53 mm. Is this an error? Is this the error that was mentioned in the abstract?

We modified "penetration depth" to read "navigation depth" everywhere in the document.

The error is the one mentioned in the abstract. To clarify what was computed, we have changed the language to read: "The average (mean) error for the maximum navigation depth was found to be 0.53mm." [R1-3]

- "... using an external motorised drive system (Fig. 1B)" readers may expect to see the motorised drive system in Fig. 1B.

We agree with the reviewer, we did not realize this in the first instance. We added a representation of the drive system in Fig. 1C and modified the text accordingly as indicated by label [R1-4].

- Can the authors please comment on the safety of the temperature increase in a human body; they mention it will be higher than that of the phantom, but no safety discussion around this is presented.

Heating would be higher in soft tissue, as the optical attenuation at 1064 nm is greater than in the phantoms used in this study. Bulk localized heating from these exposures in tissue would need to be monitored, but are not likely a significant issue. In fact, it is the increased optical absorption/heating provided by the plasmonic gold nanorods that ensures a thermal dose sufficient for thermal ablation. For comparison, current clinical uses of laser ablation for cancer have power levels ranging from 2-30W for continuous wave lasers (*Schena E, Saccomandi P, Fong Y. Laser ablation for cancer: past, present and future. Journal of functional biomaterials. 2017 Jun 14;8(2):19.*), in this study our output was set to 0.5W. Please find safety clarifications labelled [R1-5].

- More details need to be provided regarding the laser exposure. It is not clear what the distance from the fibre output end to the tissue-mimicking phantom is. Is this distance reasonable in surgery? This approach assumes the tumour is in front of the fibre. Can the laser energy be directed to a tumour on the side of the bronchi? These points need to be clarified to ensure the feasibility of this approach.

Our apologies that this value was not clear, a distance of 20 mm was used between the fiber output and the phantom during testing. This distance was selected to evaluate the heating while allowing for monitoring using the thermal camera. In surgery, the fiber end would be likely be closer to the lung wall as you would want to maximize the optical energy delivery into the target site. Refinement of this process and modification of the fiber tip (e.g. to include steering optics) would be the next phase of this application. As demonstrated here, navigation of the tentacle to the local area was the first key challenge.

We highlighted the section of the manuscript which originally stated the details of the phantom positioning, as well as adding language to clarify the steering of the laser. These are labelled as [R1-6].

- Please check the format of the references

We thank the reviewer for the suggestion. We found a few citations where the format was not consistent and made the appropriate changes.

Although existing magnetic catheter-based robotic technologies show great flexibility and have been demonstrated in different clinical applications, this paper is comprehensive about the use of magnetic tentacles integrated with shape sensing and therapy components for the treatment of bronchial tumours. The work will be interesting to the engineers in the medical robotics field and to the general readership of Communications Engineering.

We thank the reviewer for recognizing the value of our work and hope we made the appropriate changes, in response to their comments.

Reviewer #2

This is a very interesting paper that demonstrates the magnetic actuation of a tentacle to deliver laser ablation in multiple locations of the lung. Several major claims are made including the ability of the tentacle to navigate autonomously and without deformation of the surrounding tissue via feedback from Fiber Bragg Grating sensors. Optimization of the magnetic actuation methods allow for more than one pathway so a single tentacle can address multiple target locations. The clinical goal is to treat the target cells using laser ablation techniques. This paper will be of great interest to the community

because of its contributions to soft, medical robotics. The introduction of autonomous magnetic navigation to multiple target areas using 4 degrees of freedom and the integration of therapeutic tools into a miniaturized robot are contributions with merit. This paper will help advance the field and influence the technological capabilities of soft robots in minimally invasive surgical applications. The methods used by the author result in a novel contribution to the field. However, further evidence is required to strengthen the conclusions of this work as detailed in the questions below.

We thank the reviewer for their constructive feedback. We are glad that they found the paper of interest, and have endeavored to clarify the points raised in the following review.

In general, the use and safety of lasers in bronchoscopy may be further explained in the Introduction. The authors discuss only a few state of the art robots for bronchoscopy and magnetically actuated robots. To frame the work more effectively in the field, it is recommended that the authors expand on the review of relevant and recent works. A further review of laser surgery, particularly with robotics, soft magnetic robots, and other robots developed for bronchoscopic procedures would strengthen the novelty claimed in the paper. Some starting points for references on state-of-the-art robots in bronchoscopy and the use of lasers in surgery are:

1. York, P. A., Peña, R., Kent, D., Wood, R. J. (2021). Microrobotic laser steering for minimally invasive surgery. *Science Robotics*, 6(50), 5476.
2. Lee, H.C., Pacheco, N.E., Fichera, L., Russo, S. (2022). When the End Effector Is a Laser: A Review of Robotics in Laser Surgery. *Advanced Intelligent Systems*, 4(10), p.2200130.
3. Swaney, P. J., Mahoney, A. W., Hartley, B. I., Ramirez, A. A., Lamers, E., Feins, R. H., Alterovitz, R., Webster, R. J. (2017). Toward Transoral Peripheral Lung Access: Combining Continuum Robots and Steerable Needles. *Journal of Medical Robotics Research*, 2(1).
4. Van Lewen, D., Janke, T., Lee, H., Austin, R., Billatos, E., Russo, S. (2023). A Millimeter-Scale Soft Robot for Tissue Biopsy Procedures. *Advanced Intelligent Systems*, 2200326.
5. Wang, L., Zheng, D., Harker, P., Patel, A. B., Guo, C. F., Zhao, X. (2021). Evolutionary design of magnetic soft continuum robots. *Proceedings of the National Academy of Sciences of the United States of America*, 118(21), e2021922118.

We added the identified references as follows:

- 1 & 2 added in the introduction when discussing the combination of robotics and laser therapy.
- 3 & 4 added in the introduction when discussing different robotic platforms.
- 5 added to the conclusions.

These changes reflect in the original manuscript and are labelled with **[R2-1]**.

One of the major claims of the paper is the ability of the tentacle to conform its shape at multiple points along its length so that the tentacle is non-disruptive to the lung walls. However, it seems that there are instances when the sensor detects that the tentacle is outside of the lung which implies that it is contacting the lung walls. In the phantom navigation experiments, the tentacle looks as if it is contacting the lung to make turns (Navigation B, Movie S1). The potential deformations due to this contact should be quantified to strengthen the claim of full shape control. The methodology employed for minimizing contact should also be explained as this is unclear. Contact minimization is claimed in line 450 and both Cadaver and Phantom Experiments in the paper but not explained in detail. Similarly, the localization error was stated to be 5 mm which is significant at smaller lung branches in the periphery. Does this localization error affect shape control?

We apologize if this was not made clear in the original manuscript. The main goal of the proposed approach is not to eliminate contact with the anatomy, which is an almost impossible task. Our aim is to be able to guarantee navigation, without the need for functional contact. In fact, in most cases of magnetic catheters with tip and/or axial magnetization, contact is necessary to navigate. Our approach uses a follow-the-leader implementation which guarantees that, ideally, the tentacle shapes as desired even without contact. Thus, it does not require contact to navigate. We have made this concept clearer in the updated version of the manuscript and labelled it with [R2-2].

We would also like to underline that, we do not claim to entirely eliminate contact, but rather eliminate the need for contact to achieve successful navigation. Due to the softness of the tentacle, interactions with the anatomy identified in the localization will be inherently limited in force, as can be seen in the comparison with a standard catheter in Figure X - reducing safety concerns.

In the proposed paper, shape sensing is used to provide feedback to the clinician on the location and shape of the tentacle. This is used to reduce x-ray exposure, which is important for both staff and patient. Closed-loop control was not used at this stage, since we first wanted to investigate the ability to navigate in open loop. Therefore, the clinician is left to interpret the localization results, which are used as a guidance to know in which branch the tentacle has navigated to. We found that this error does not create problems in interpreting the tentacle's location, given the branches of the bronchi are far enough and the clinician can merge this information with bronchoscope's camera view.

In the future, when closed loop is investigated, we will consider improving registration (pre- to intra-operative image alignment), segmentation (bronchial tree extraction from CT) and localization. These factors are expected to improve the overall outcome if we aim at limiting the presence of the clinician in the loop.

Magnetic actuation has the significant benefit of controlling multiple degrees of freedom simultaneously with the same set of EPMS. How does the design of the tentacle such as spacing, length, and amount magnetic segments affect its ability to conform to pathways? Further comments on these design choices would be helpful.

We decided on the number of segments given the type of pathways we face. Study on optimization of these parameters, as in the paper "Evolutionary design of magnetic soft continuum robots.", could be addressed in future work. We mention this in the discussions and label it with [R2-3].

The authors state that optimization was performed for each of 8 navigations to produce the optimal magnetization profiles for each navigation. After this, the 8 different fields to achieve each navigation was derived for each of the 8 magnetization profiles. The optimization takes the magnetization profile with the minimum error. Is it possible there is a more optimal magnetization profile that is not considered as a result of this process? Can the magnetization profile be optimized once to all 8 navigations rather than just each individual navigation?

This is an interesting query and the authors are currently investigating options for creating magnetization profiles which are globally optimal for a range of geometries. This makes clinical sense in the case of, for example, navigating through dynamic environments such as breathing lungs. We do, however, consider this development out of scope for the current contribution.

The optimization for any given navigation is 24 dimensional. If we were to optimize for all navigations simultaneously, we would be dealing with a very high dimension problem with its own inherent challenges. Going forward, the ambition is to automate fabrication such that geometry/patient specific bespoke catheters are manufactured and magnetized for each independent navigation. Thus, the range of geometries (e.g. peak to trough of a respiratory cycle) which any unique design will need to conform to will be far lower than that which we are attempting here.

Furthermore, for these (static geometry) navigations we have tried to keep our theoretical design methodology as consistent as possible with previous publications (13) such that the progression of our catheter design follows a coherent narrative thread as we add complication in other areas (fabrication, localization and clinical demonstration).

In light of these factors, we have modified the discussions and labelled it with [R2-4].

This question pertains to the experimental setups and the strength of the EPMs. What about the case when there is additional anatomy between the lungs and the EPMs as is the case in a real clinical setting? If the distance from the EPMs to the tentacle required for navigation is less than is allowed due to the anatomy, will there be any method to account for this?

At the moment we impose a maximum distance of approximately 50 cm between the EPMs, which fits the anatomy and is seen to be acceptable in the cadaveric experiments. We envision that, an increase in the magnets size will be required to fit any possible patient. The generated field would scale in a known fashion and may require utilization of robots able to carry higher payload. We believe this is an important development to be made, however current research priority is being placed on development of the tentacle technology itself. Their stiffness and magnetic content (under investigation) would change the requirements in the generated field and impose specific constraints.

We acknowledge this in the comment labelled [R2-5], in the discussions.

The authors state that the location of the tip relative to the tip of the bronchoscope is determined based on the speed at which the tentacle is advanced by a stepper motor. However, it is also mentioned that there is a pulling force imposed on the magnetic segments by the EPMs. How does the pulling force effect the speed and, consequently, the location of the tip? Potential contact interactions with the lung walls should also be considered as this can affect the speed and direction of the tentacle.

The pulling force does not override the insertion step of the drive system nor does the interaction with the anatomy, since is only strong enough to affect the catheter and not the motor at the base of the drive system. The interaction with the anatomy is indeed considered, as measured by the FBG. The FBG and laser makes it stiff in extension.

Laser ablation is a therapeutic technique that has significant potential in treating these types of lung cancers. There are a few questions on how it is used in this paper. The authors claim that the laser is highly localized. However, the area affected by the laser seems large in Fig. 3 relative to the scale of the lung anatomy. Can the tentacle navigation be used to control the size of the affected area? Are there any unexpected losses in laser power as a result of the fiber being bent in the anatomy? Will the laser cause heating of the tentacle and affect the sensitivity of the magnets or the shape sensing FBG?

This is a good point. However, the distance between the output of the fibre and phantom (20 mm) was chosen specifically to allow for monitoring with the thermal camera and would not be expected to replicate clinical conditions. Furthermore, since the laser field would be diffusive after a few mm into lung tissue, the tentacle would only need to navigate to the local area. This is why having the nanorods present in the tumour is important, as they can be molecular targeted and systematically introduced (around 24h prior to surgery) they increase the optical absorption in this region causing greater heating than normal or healthy tissue. Accurate tentacle navigation is important to ensure that the laser is directed to the correct local area, and the illumination angle and spot size of the beam on the lung wall will affect how the beam diffuses when propagating through tissue.

Bending losses could be significant if the tentacle exceeds the bend radius of the fibre. For the ones used in this study this was specified as 120 x Cladding Diameter (125um), so approximately 15mm, which is less than any bends needed for navigation.

Since fibre optics are very efficient at transporting light along their length. The total attenuation loss (a percentage would convert to heat, depending on the optical absorption properties of the surrounding material) was stated at 8dB/km, and given the total length of a fibre was 2 m this represents a power loss of 0.16dB. This would correspond to a power loss of 22.5 mW over the length of the fibre, likely resulting in a negligible heat rise in the fibre cladding/coating. However, if higher powers were used, or a less efficient material was used this could be higher and may pose an issue, which would need further investigation.

We have added language in the paper, to clarify the points raised by the reviewer and labelled them as [R2-7].

In a clinical situation, the target cells will likely not be directly in front of the robot but rather along the walls of the lungs. The laser ablation was only demonstrated directly in front of the tentacle. Is the tentacle able to steer the laser so that it can aim towards the lung walls and achieve a close enough distance to effectively ablate the cells? Further, can this maneuver be optimized so that the robot minimizes contact with the lung walls as is claimed?

In the clinical scenario, the fiber end would likely be closer to the lung wall compared to the experiments, guaranteeing higher heating at lower laser power. We will work on refinement of the fiber tip (e.g. to include steering optics) to guarantee targeted ablation of the surrounding tissue. Please, see additional text in the new version of the manuscript labelled as [R2-8].

In the Cadaveric Experiments, analysis of the repeatability is a difference between maximum penetration in each view of the fluoroscopic images. It is unclear whether the mean value reported is in a specific view or across all views in the fluoroscopic images. Further, reporting a standard deviation for this analysis would strengthen the claim of repeatability and prove that the mean value can be reliably repeated.

We made this clearer in the new version of the manuscript and added standard deviation. This is labelled as [R2-9].

Does the approximation error of the second order polynomial compound with any sensor error that is attributed to limited resolution? The resolution of the FBG sensor is claimed to be a source of error. However, its accuracy is not analyzed. What is the error of the FBG sensor? More details on the calibration method would likely help with this explanation as well.

The accuracy of the FBG sensor depends on different factors such as, reference point chosen, the bending radii of the application, potential twist and sensor length. The typical tip position error for a 200 mm sensing length is about 1.2%. Despite the error attributed to the approximation error of the second order polynomial, the FBG's sensitivity, and that of the reference point (whose position was obtained using the OptiTrack system), we believe that a large percentage of the error recorded is due to the tentacle buckling or slipping, and discrepancies in the registration between the scan and the cadaveric specimen.

We added language to the manuscript (label [R2-10]) to clarify the calibration/registration errors.

Reviewer #3

The authors present a system for targeted photothermal cancer therapy in the respiratory system using a flexible magnetic guide wire containing a laser fiber for local heating and a FBG fiber for shape sensing. The “magnetic tentacle” is steered into several lung branches using two external permanent magnets. The aim of this study is clear and the shown experiments demonstrate the capabilities of this method well. The paper is well-written. The validation of the method in a cadaver lung is good.

However, the referee feel that several major points should still be clarified before publishing. My detailed comments are below.

We thank the reviewer for their constructive feedback, and we are glad they recognised the value of our work. We have modified the manuscript to implement the comments below.

Major comments:

1. The experiment for Fig. 2 (Movie S1) is unrealistic as the physical boundaries of a human torso are not respected. The two external magnets move freely in space and do not appear to have constrains in their movement. Realistic orientations and distances of the external magnets in relation to the bronchial tree should be considered.

We understand the reviewer's concern and, indeed the magnets might transition between poses violating the constraints. This is because we have not fully addressed the safety issues yet, for the EPMs moving point-to-point. We are working on EPM path planning to respect safety constraints in future work. As it can be seen with the cadaveric experiments, the bronchial tree is inside the lungs and constrained in a box. In this case (cadaver testing), the robots are seen to never hit the fake torso. In this work, we could guarantee an EPM-EPM distance of 50 cm.

In the future, we will also consider larger magnets, whose generated magnetic field can be made significantly higher and allow the EPMs to be further away from the patient. This, however, may need to change the robotic manipulators, which were selected for ease in the development of the proposed concept. We will consider this in the future, after we optimize the magnetic and mechanical properties of the tentacles. As there is continuous development in this direction, we would first study the possibility of applying lower fields via more optimal fabrication (magnetic content and stiffness).

In the manuscript, we acknowledge this in the discussions. **[R3-1]**

2. The localization error (Table 1 and 2) is not well defined. It is not clear to the referee how the spatial error is measured. Is it an average over the whole tentacle? Do the authors take the length of the tentacle into account to normalize the error? A clear mathematical definition is necessary.

The error presented in Tables 1 and 2 represents the percentage of tentacle within the 3D mesh of the anatomy at every time step, averaged over the total insertion time. This can be expressed by the equation:

$$err = mean_{t \in [0, T]} \frac{p_{in}(t)}{p_{out}(t)} \cdot 100$$

with $p_{in}(t)$ number of points along the length of FBG insider the anatomy, $p_{out}(t)$ number of points outside the anatomy and T final time. We added this to the revised version of the paper and labelled it with **[R3-2]**.

3. Fig. 3 illustrates the photothermal delivery results on phantom. It is very confusing. Does the material mimic tissue property for laser absorption? Where do the gold nanoparticles come from, and for what purpose? Is the result relevant to the tentacle navigation and localization? If the authors believe it is highly relevant, this should be shown in conjunction with the segmental bronchi phantom.

We apologize for the confusion. We demonstrate laser delivery in the bronchi, which can be used for curative purpose. To improve targeting, functionalized gold nanoparticles able to bind to specific tumoural cells in the lungs can be injected in the blood stream, as shown in literature. Please, see comments labelled as **[R3-3]**.

4. The overall flow of the manuscript needs careful consideration and modification. The referee believe it is easier for the readers to follow if the tentacles structure, fabrication, and actuation (Fig. 5-7) are discussed earlier, before the navigation results (Fig. 2 and Fig. 4).

We understand the reviewer's concern and, in our field, we generally have the methods first. The format of the paper we propose here is somewhat different from what we are used to, however, we approached it to best fit the format required by the journal and we have contacted the EiC to understand the strictness of the paper's organization. Accordingly, the EiC would rather have us keep this format for the journal, to respect the guidelines.

5. Why is an extended Kalman filter applied to determine the position of the bronchoscope? How was the error determined? How long did the calibration of the position and orientation take?

In order to localize the tip of the bronchoscope with respect to the robots and pre-computed anatomy, we used a localization method based on Hall effect sensors. The need for accurate knowledge of the bronchoscope's tip is twofold: the robots need to be informed of where the field generated has to focus; the surgeon is to be informed of where in the anatomy the tentacle is navigating. The latter is achieved by combining the information from the Hall effect sensor (through filtering with EKF to find the position) and the measurement of the shape from FBGs. The latter alone can only provide information on how the tentacle is shaping but no information on where it is.

The error was determined by testing the localisation method against optical tracking data (see comment below).

The calibration of the position and orientation took around 300 seconds to complete. It involved moving the EPM around the workspace for over 200 seconds and then generating three magnetic fields in the three inertial directions.

We added language to explain these concepts in the section "Localization and Shape Sensing" as well as a citation to our most recent paper on the topic (28) that has recently been accepted for publication in IEEE RAL, but available to the reviewer and readers via arxiv. These changes are labelled as [R3-4].

6. What is the physical accuracy of the new localization method? Can the authors give quantitative evaluation about this on a regular spatial grid as a ground truth? How much does the localization error depending on the magnetic localization or the tip of the bronchoscope?

We have added a citation to our most recent paper on the topic, where we discuss the magnetic localization, here applied to the tip of the bronchoscope, in the details. We have not provided additional experiments here, since the paper (28) has covered all the aspects the reviewer is referring to and we avoided adding already published material. Please, consider the label of the previous comment's changes for additional language on this.

7. Does the permanent magnetic actuation affect the magnetic localization? How was this problem solved?

The permanent magnets do indeed play an important role in the localization, since they are the source of field used to localize the tip of the bronchoscope. Their presence does not cause problems, on the contrary, they provide a solution to the localization of the overall system. We underlined the section of the paper where this is expressed with the label [R3-5].

Additional comments:

1. The figures have inconsistent quality with respect to readability, font size, dimensions and overlays.

We have made all the figures more homogeneous in terms of font and size.

2. The abstract is too long and too detailed. Main novelties should be summarized concisely.

We have shortened the abstract and focused it on the novelties of the presented work. Please, see the label [R3-6].

3. Fig. 2:

a. The image quality is quite low

We printed the phantom used in the most transparent material we could. Therefore, we could not produce images of higher quality, unfortunately.

b. Why do the authors not show the whole image and mark the relevant area separately?

We tried the approach proposed by the reviewer, but overlapping all those images would produce a very unreadable figure.

c. What is the artifact in D on the right side?

The artifact is a shadow created by the EPM positioned behind the phantom. Due to the transparency of the phantom's material the EPM partially appears and is visible through the anatomy.

4. Movie S1: It would be beneficial to pause the video for a few seconds when the target is reached.

We agree with the reviewer on this point, and we stopped the video at the end of each navigation for 5s.

5. Fig. 5: The figure labels do not read left-to-right and up-to-down.

The reason for this choice is that we preferred ordering the figures so that there would be limited empty space in the figure, so to optimize its size in the paper. We found that the order of the labels is more useful because they follow the discussion in the manuscript.

6. What is the grade/remanence field of the magnetic particles?

The remanence is 0.903 T. We added this detail in the manuscript and labelled it with [R3-7].

7. Spelling mistake: Mahony filter

8. Reference to the Mahony filter is missing

We corrected the spelling mistake and added the reference. Please, check label [R3-8].

Reviewers' comments:

Reviewer #1 (Remarks to the Author):

Many thanks to the authors who have responded to most of my comments. The following items still need attention.

R1-2: The error, as defined, will increase with the increase of P_{in} . Can the authors please check the formula? Authors may also consider using the total number of points in the denominator for the error calculation.

R1-6: The authors still mention the distance from the exit of the bronchi, not from fibre output. Are these the same? They also mention laser-phantom distance, which would be clearer if fibre output to phantom distance were used instead, as they did in their rebuttal.

Reviewer #2 (Remarks to the Author):

The authors have done a good job addressing this reviewer's comments.

A couple of minor comments below.

1. Open-loop control in a clinical setting could lead to some unwanted behavior during navigation and/or interactions with lung tissue due to errors in sensor readings and initial localization. This is not made clear in the paper as the only control strategy mentioned is supervised autonomy.
2. It should be clarified that this paper implements open-loop navigation and that shape sensing is only feedback to the clinician and not the robot itself. The wording in some of the sentences made this unclear.

Reviewer #3 (Remarks to the Author):

The authors have majorly revised the manuscript following all referees' comments. In this referee's opinion, the paper can be published at the current state.

Revision of COMMS-ENG-23-0016-A

We thank the editors and reviewers for their constructive feedback on the paper's form and content. We gladly applied all changes requested and responded to all the comments, as follows in the present document.

In the following, our response was highlighted with **red text**. In the revised manuscript:

- **Removed text** indicates major reduction in the text.
- **Red text** indicates additional language, in response to reviewers' comments.
- **Blue text** indicates text of the original manuscript which we highlighted in response to a specific comment from the reviewers.
- **[Ri-j*]** labels where the reviewers can find changes reflecting the comment **j** of the reviewer **i**.

Reviewer #1

Many thanks to the authors who have responded to most of my comments. The following items still need attention.

R1-2: The error, as defined, will increase with the increase of P_{in} . Can the authors please check the formula? Authors may also consider using the total number of points in the denominator for the error calculation.

We apologize for the confusion. The actual equation is

$$err = \text{mean}_{t \in [0, T]} \frac{p_{out}(t)}{p_{in}(t)} \cdot 100$$

with $p_{in}(t)$ total number of points inserted as we report in the new version of manuscript and labelled as [R1-1*]. In the previous version, there was a mistake in numerator/denominator and description.

R1-6: The authors still mention the distance from the exit of the bronchi, not from fibre output. Are these the same? They also mention laser-phantom distance, which would be clearer if fibre output to phantom distance were used instead, as they did in their rebuttal.

We apologize for the confusion. Yes, the end of the phantom and the output of the laser fibre are coincident. We clarified this in the new version of the manuscript and labelled as [R1-2*].

Reviewer #2

The authors have done a good job addressing this reviewer's comments.

A couple of minor comments below.

1. Open-loop control in a clinical setting could lead to some unwanted behavior during navigation and/or interactions with lung tissue due to errors in sensor readings and initial localization. This is not made clear in the paper as the only control strategy mentioned is supervised autonomy.

We understand the concern raised by the reviewer. We clarified that autonomy is achieved in open loop using only pre-operative imaging and optimization in the introduction. In the discussion, we made clearer what the consequences of open loop control may be. These changes were labelled as [R2-1*].

2. It should be clarified that this paper implements open-loop navigation and that shape sensing is only feedback to the clinician and not the robot itself. The wording in some of the sentences made this unclear.

We added clarifying wording in the introduction. We also added a sentence on how closed loop control will be implemented in future work. Both changes are labelled as [R2-2*].

Reviewer #3

The authors have majorly revised the manuscript following all referees' comments. In this referee's opinion, the paper can be published at the current state.

We thank the reviewer for their work on the manuscript and we are glad that the value of our work was recognised.

REVIEWERS' COMMENTS:

Reviewer #1 (Remarks to the Author):

The authors made all the corrections I asked for. I recommend the publication of the manuscript as it is. Many thanks.

Reviewer #2 (Remarks to the Author):

The reviewers have addressed all necessary comments and edits. The paper can be published in its current form.